

**Evaluating WRF-GC v2.0 predictions of boundary layer and vertical ozone**
**profiles during the 2021 TRACER-AQ campaign in Houston, Texas**
**Xueying Liu[1], Yuxuan Wang[1], Shailaja Wasti[1], Wei Li[1], Ehsan Soleimanian[1], James**
**Flynn[1], Travis Griggs[1], Sergio Alvarez[1], John T. Sullivan[2], Maurice Roots[3], Laurence**
**Twigg[4], Guillaume Gronoff[5], Timothy Berkoff[5], Paul Walter[6], Mark Estes[6], Johnathan W.**
**Hair[5], Taylor Shingler[5], Amy Jo Scarino[4], Marta Fenn[4], Laura Judd[5]**
[1]Department of Earth and Atmospheric Sciences, University of Houston, Houston, TX, USA
[2]NASA Goddard Space Flight Center, Greenbelt, MD, USA
[3]Department of Physics, University of Maryland Baltimore County, MD, USA
[4]Science Systems and Applications, Inc., Hampton, VA, USA
[5]NASA Langley Research Center, Hampton, VA, USA
[6]School of Natural Sciences, St. Edward's University, Austin, TX, USA
**Correspondence:** Yuxuan Wang (ywang246@central.uh.edu)
**Abstract.** The Tracking Aerosol Convection Experiment Air Quality (TRACER-AQ) campaign
probed Houston air quality with a comprehensive suite of ground-based and airborne remote
sensing measurements during the intensive operating period in September 2021. Two post-frontal
high-ozone episodes (September 6-11 and 23-26) were recorded during the said period. In this
study, we evaluated the simulation of the planetary boundary layer (PBL) height and the vertical
ozone profile by a high-resolution (1.33 km) 3-D photochemical model, Weather Research and
Forecasting (WRF)-driven GEOS-Chem (WRF-GC). We contrasted the model performance
between ozone-episode days and non-episode days. The model captures the diurnal variations of
the PBL during ozone episodes (R = 0.72-0.77; normal mean bias (NMB) = 3%-22%) and non-
episode days (R = 0.88; NMB = -21%), compared with the ceilometer at La Porte. Land-water
differences in PBL heights are captured better during non-episode days than episode days,
compared with the airborne High Spectral Resolution Lidar-2 (HSRL-2). During ozone episodes,
the simulated land-water differences are 50-60 m (morning), 320-520 m (noon), and 440-560 m
(afternoon) in comparison with the observed values of 190 m, 130 m, and 260 m, respectively.
During non-episode days, the simulated land-water differences are 140-220 m (morning) and 360-
760 m (noon) in comparison with the observed values of 210 m and 420 m, respectively. For
vertical ozone distributions, the model was evaluated against vertical profile measurements from
the Tropospheric Ozone lidar (TROPOZ), the HSRL-2, and ozonesondes, as well as at the surface
from a model 49i ozone analyzer and a site from the Continuous Ambient Monitoring Stations
(CAMS) at La Porte. The model underestimates free tropospheric ozone (2-3 km aloft) by 9%-22%
but overestimates near-ground ozone (< 50 m aloft) by 6%-39% during the two ozone episodes.
Boundary layer ozone (0.5-1 km aloft) is underestimated by 1%-11% during September 8-11 but
overestimated by 0%-7% during September 23-26. Based on these evaluations, we identified two
model limitations: the single-layer PBL representation and free tropospheric ozone
underestimation. These limitations have implications for the predictivity of ozone's vertical
mixing and distribution in other models.





## 1. Introduction

The Houston metropolitan area has experienced nonattainment of the US National Ambient Air Quality Standards (NAAQS) for ozone over decades (TCEQ, 2022). Ozone exceedances in Houston usually occur in two peaks, a spring peak in April–May and a late summer peak in August–October (Zhou et al., 2014). Such seasonal behavior is driven by diverse meteorological conditions that influence ozone development. The passages of synoptic-scale cold fronts (~ 1000 km horizontally and ~ 5 km vertically; a timescale of days) are known to bring high background ozone air from the continent into the Houston area (Lefer et al., 2010; McMillan et al., 2010; Haman et al., 2014). Mesoscale sea breeze recirculation (~ 20 km horizontally and ~ 1 km vertically; a timescale of hours) is found to be associated with ozone exceedances (Li et al., 2020; Banta et al., 2005, 2011; Caicedo et al., 2019). Meanwhile, microscale-to-mesoscale vertical mixing (< 1 km vertically; a timescale of hours) of the lower troposphere is shown to be a significant factor in near-surface ozone air quality (Morris et al., 2010; Haman et al., 2014; Sullivan et al., 2017; Xu et al., 2018; Caputi et al., 2019). Favored by these meteorological conditions of different scales, local emissions of ozone precursors from the urban center and the nearby Houston Ship Channel stay locally in the area and lead to high-ozone events. This study will focus on the impact of mixing between lower free tropospheric layers on vertical ozone distribution and the impact of chemistry is outside the scope of this analysis.

The planetary boundary layer (PBL) is the lower part (e.g. < 2 km) of the troposphere that is directly influenced by the presence of the Earth's surface and responds to surface forcings with a timescale of an hour or less. A stable capping layer at the top of the PBL, where temperature increases with height, is known as the capping inversion (CI) layer (e.g. ~ 2 km). With the cap in place, air exchange is inhibited between the overlying free troposphere (FT) (e.g. >2 km) and the underlying PBL (e.g. < 2 km). During the daytime, there is strong turbulence production throughout the PBL, generating a buoyant layer called the convective boundary layer (CBL). The CBL is characterized by intense mixing in a statically unstable situation where warm air rises from the ground, growing from a few hundred meters in the early morning (e.g. ~ 0.5 km) towards the top of the PBL in the afternoon (e.g. ~ 2 km). As the sun sets, convectively driven turbulence decays in the formerly well-mixed CBL. The remnant of the recently decayed CBL will remain aloft in the less-turbulent residual layer (RL) at around 1–2 km. As the night progresses, the bottom portion of the RL transforms into a stable boundary layer (SBL) (e.g. < 0.5 km) due to its contact with the ground, characterized by statically stable air with weak and sporadic turbulence. The PBL is commonly considered as the CBL under certain conditions during the daytime and the SBL during the nighttime (Tangborn et al., 2021).

The heights of the PBL (incl. CBL and SBL) and other lower tropospheric layers (e.g. RL, CI) are defined mainly by temperature inversions. It is primarily a thermodynamic-based definition. Atmospheric models adopt the thermodynamic concept and rely on parameterization schemes to define the structure of the PBL and compute the height of the PBL. Two major types of data have been commonly used to validate the modeled PBL height. The first type is the PBL height derived from the profiles of thermodynamic properties measured by ozonesondes and radiosondes (Zhang et al., 2019; Zhang et al., 2020; Morris et al., 2010; Rappenglück et al., 2008). These observations share a similar thermodynamic definition with the modeled PBL height and are widely used to validate model prediction of the PBL height under various conditions (day, night, land, water). The second type of data is remotely-sensed mixed layer



height as defined by aerosol backscatter gradients, which is becoming more widely available
with ceilometer data and aircraft lidars and can be adopted for model evaluation (Caicedo et al.,
2017, 2020; Knepp et al., 2017; Li et al., 2021; Wang et al., 2020).
Mixed layer height, defined as the volume of atmosphere in which aerosols are well mixed and
dispersed, can be derived from the unattenuated backscatter signal of aerosols alone (e.g. the
High Spectral Resolution Lidar-2 (HSRL-2)) or the attenuated total backscatter signal produced
by aerosols and molecules combined (e.g. CHM 15k-x ceilometers). Both signals have been used
to derive mixed layer height for model comparisons (Scarino et al., 2014; Li et al., 2022). Mixed
layer height does not equal PBL height by definition; it approximates the CBL height during the
daytime and can represent the height of the RL or the SBL depending on retrieval algorithms
applied to lidar signals at night (Wang et al., 2020; Vivone et al., 2021). Mixed layer height is
often a good proxy for the heights of different lower tropospheric layers determined
thermodynamically in models during the daytime (Scarino et al., 2014) and throughout the day
(Kuik et al., 2016; Haman et al., 2014) and serves as an input parameter of PBL heights for
meteorological and photochemical models (Tangborn et al., 2021; Knote et al., 2015; Geiß et al.,
16 2017).

Vertical mixing between different layers of the lower troposphere, such as boundary layer
mixing with the FT flow at its upper interface (through entrainment processes), mixing between
the RL and the SBL (through surface exchange processes) and the RL mixing through the growth
of the CBL, etc., strongly influences surface ozone concentrations. Entrainment can occur during
the daytime when strong convective thermals penetrate the laminar FT above and then sink back
into the CBL, bringing the FT air towards the surface and thus affecting surface ozone
concentrations (Parrish et al., 2010; Jaffe et al., 2011). Located between the FT and the CBL, the
strength of the CI layer limits the upward penetration of thermals and is thus used to indicate the
influence of the FT air on surface ozone (Kaser et al., 2017; Morris et al., 2010; Rappenglück et
al., 2008). Meanwhile, surface exchange processes occur when a low-level jet exists between the
RL and the underlying SBL and drives the shear production of turbulence between these layers.
Since the RL is a known ozone reservoir with limited $NO_x$ titration and ozone deposition, ozone-
rich air in the RL can be mixed down into the SBL effectively, where it is subject to dry
deposition to the surface, affecting surface ozone concentrations (Tucker et al., 2010; Sullivan et
al., 2017; Caputi et al., 2019; Bernier et al., 2019; Zhao et al., 2022; Xu et al., 2018).
The Tracking Aerosol Convection Experiment Air Quality (TRACER-AQ, https://www-
air.larc.nasa.gov/missions/tracer-aq/) campaign, led by NASA with contributions from the Texas
Commission on Environmental Quality (TCEQ), probed Houston air quality with a
comprehensive suite of remote sensing and in situ measurements of ozone, ozone precursors, and
meteorology from ground-based, airborne, balloon-borne and shipborne platforms (Jensen et al.,
2022). The operational period occurred from July–September 2021, with intensive measurements
during September 2021. Combining field campaign observations with a high-resolution 3-D
photochemical model, the goals of this study are to (1) evaluate the PBL prediction in the model,
(2) examine the vertical distribution of ozone, and (3) identify specific model limitations that
prevent accurate prediction of the PBL height and the vertical ozone distribution.



**2. Model and Data**
**2.1 Observations**
To evaluate the PBL and the vertical ozone distribution, this study adopted continuous, high-
resolution profiles (i.e., 1–10 minutes) from ground-based measurements at the La Porte site and
airborne measurements covering urban Houston and the Galveston Bay in September of 2021
(Fig. 1). Compared with discrete or low-resolution PBL measurements (*e.g.* hourly) used in
previous studies in Houston (Haman et al., 2014; Cuchiara et al., 2014; Rappenglück et al.,
2008), the high-resolution measurements in TRACER-AQ field campaign are capable to probe
into the fine PBL structure and its development as well as the associated vertical ozone profiles.
The La Porte site was equipped with (1) semi-continuous vertical ozone profiles from the NASA
Goddard Space Flight Center (GSFC) Tropospheric Ozone (TROPOZ) Differential Absorption
Lidar (DIAL) (Sullivan et al., 2014), (2) continuous aerosol mixed layer height derived from
atmospheric backscatter profiling with a CHM 15k-x ceilometer, (3) multiple ozonesonde
launches, and (4) continuous surface ozone and meteorology measurements.
The TROPOZ, as part of the ground-based Tropospheric Ozone Lidar Network (TOLNet,
https://www-air.larc.nasa.gov/missions/TOLNet/), has been used to provide continuous, high-
resolution profile measurements of vertical ozone profile during various campaigns for satellite
and model evaluation (Sullivan et al., 2014, 2015, 2019, 2022; Bernier et al., 2022; Kotsakis et
al., 2022; Dacic et al., 2020; Johnson et al., 2016; Dreessen et al., 2016). The TROPOZ data can
be used to identify pollutant transport to understand the vertical mixing of ozone. Similar to the
TROPOZ at the La Porte site, the University of Houston site measured semi-continuous vertical
ozone profiles with the Langley Mobile Ozone Lidar (LMOL) (Gronoff et al., 2019, 2021).
The CHM 15k-x ceilometer measured continuous atmospheric attenuated backscatter profiles at
a wavelength of 1064 nm. The signals were corrected due to the incomplete superposition of the
laser and the receiver field of view by the overlapping correction function from the manufacturer
(Rizza et al., 2017). The normalized range corrected signals (RCS) is shown in this paper. The
sharp gradients in the collected backscatter were then used to detect up to three aerosol layers by
the standard retrieval algorithm provided by the ceilometer manufacturer (Lufft, 2016). The
lowest determined aerosol layer is characterized as mixed layer height. It depends on the users to
determine whether the derived mixed layer height can be used as a proxy for thermodynamically-
defined layers such as the CBL, the SBL and the RL (Caicedo et al., 2017, 2020; Knepp et al.,
2017; Li et al., 2021; Wang et al., 2020).
Ozonesondes were often launched multiple times in a day at several locations and measured
vertical profiles of ozone and meteorological variables including temperature, humidity, and
winds. This study uses ozone and potential temperature profiles from eight ozonesondes at La
Porte launched on 10:00-15:00 CDT during ozone episodes.
Surface measurements at La Porte included ozone, air temperature, relative humidity, and wind
speed and direction. This study uses surface ozone from a Thermo Scientific model 49i ozone
analyzer operated by the GSFC and a TCEQ Continuous Ambient Monitoring Stations (CAMS)
site named La Porte Sylvan Beach, as well as surface meteorology from a Lufft WS-501B
operated by the GSFC.



In September 2021, the NASA Gulfstream-V aircraft flew on ten flight days. This analysis uses
the High Spectral Resolution Lidar-2 (HSRL-2) datasets collected over the Houston area up to
three times per day, roughly at 8:00-10:00, 11:00-13:00, and 14:00-16:00 local time, covering an
area of approximately 50 km x 135 km. With its high resolution and vertically resolved
measurements, the HSRL-2 demonstrated reliable performances on many previous airborne
campaigns (Hair et al., 2018; Hair et al., 2008; Burton et al., 2015).The HSRL-2 provides below
aircraft retrievals of the spatial and vertical distributions of ozone, aerosols, aerosol optical
properties, and mixed layer heights. This paper only reports on (1) mixed layer height derived
from gradients in the aerosol backscatter profiles measured at 532 nm and (2) ozone mixing ratio
along one flight track that has the nearest distance to the La Porte site (Fig. 1).
Mixed layer heights from the ceilometer at La Porte and the HSRL-2 are derived differently. The
ceilometer at the La Porte site measures attenuated total backscatter profiles of the atmosphere
(incl. aerosols and molecules), while the HSRL-2 can measure the unattenuated aerosol
backscatter profile. Both ceilometer and the HSRL-2 signals can be used to derive mixed layer
height. This study uses mixed layer heights from the ceilometer and the HSRL-2 to evaluate the
WRF-GC prediction of PBL heights.
The supplement includes (1) surface measurements from the TCEQ CAMS and the boats
throughout July to October used to identify ozone exceedance days, (2) details on the
assimilation and evaluation of the modeled meteorology with these measurements, and (3)
vertical ozone distribution at University of Houston by LMOL.
Apart from the observations above, we used geopotential heights and winds at 850 hPa from the
European Centre for Medium Range Weather Forecast (ECMWF) reanalysis version5 (ERA5)
dataset (description in Sect. 2.3.2) to derive the synoptic conditions in Fig. 2.
**2.2 Identification of ozone episodes**
Ozone exceedance days were identified according to surface measurements from the TCEQ
CAMS (onshore) and the boats (offshore). The criteria used in this study is that any onshore site
from the CAMS network in Houston and Galveston or offshore boat ozone observations
registered daily maximum 8-hour average (MDA8) ozone in exceedance of 70 ppbv, the current
U.S. Environmental Protection Agency (EPA) NAAQS air quality standard for ozone. Three
high ozone episodes in September of 2021 were identified based on the above criteria:
September 6-11, September 17-19, and September 23-26, consisting of 13 ozone exceedance
days. We excluded analysis from the September 17-19 episode because it happened right after
tropical cyclone Nicholas, which made landfall 125 km south-southwest of Houston and
hindered measurements at the ground-sites and aircraft due to clouds and power outages. The
model meteorology was not designed to capture the cyclone either. Other September days were
used as a control to represent non-episode days.
**2.3 Model**
**2.3.1 Model description**
WRF-GC v2.0 is a regional air quality model (Feng et al., 2021; Lin et al., 2020) that couples the
Weather Research and Forecasting (WRF) meteorological model (v3.9.1.1) with the GEOS-
Chem atmospheric chemistry model (v12.7.2). The WRF and GEOS-Chem versions are



benchmarks of WRF-GC v2.0 with the proven performance of meteorology, PBL heights, and
aerosol simulation in Feng et al. (2021) and Lin et al. (2020). We evaluated WRF-GC's
prediction of ozone during the TRACER-AQ study. We set up three domains with different
horizontal resolutions that cover the contiguous United States, Southeast Texas, and the
Houston-Galveston region, referred to as d01, d02, and d03, respectively, as shown in Figure 1.
The corresponding horizontal resolutions for d01–d03 are 12 km, 4 km, and 1.33 km,
respectively. All domains have identical vertical resolutions with 50 hybrid sigma-eta vertical
levels spanning from the surface to 10 hPa. Vertical resolution ranges from ~70 m (near the
ground) to ~700 m (aloft); the first 2 km above the ground has 10 model layers, and the first 4
km has 14 model layers.

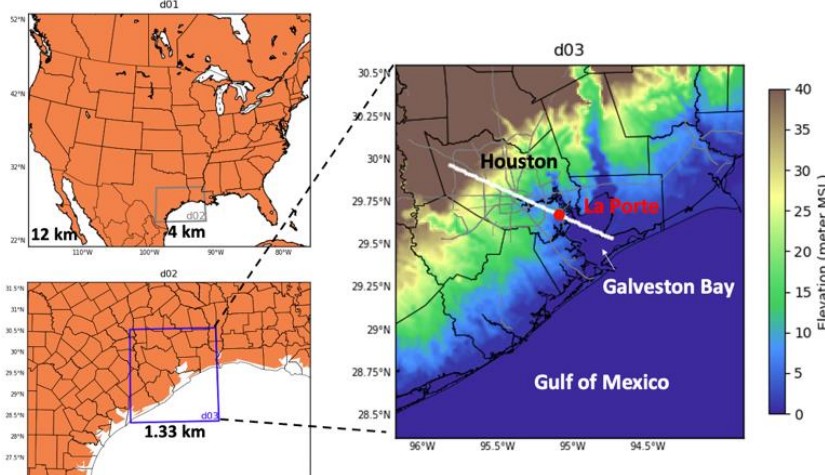

Figure 1. WRF-GC nested domains and their horizontal resolutions. The La Porte site is labeled
as a red dot. The white line represents a flight track that is chosen because of its nearest distance
to the La Porte site.
WRF-GC uses the most updated full $O_x$-$NO_x$-VOC-halogen-aerosol chemistry from GEOS-
Chem. The anthropogenic emissions used are the 2019 TCEQ emission inventory for Houston
and southeastern Texas, the 2013 National Emission Inventory for the rest of the US, and the
2014 Community Emissions Data System (CEDS) for regions outside of the US. Biomass
burning emissions are from the 2019 Global Fire Emissions Database (GFED). Biogenic
emissions are from the Model of Emissions of Gases and Aerosols from Nature (MEGAN)
(Guenther et al., 2012). Soil $NO_x$ (Hudman et al., 2012) and lightning $NO_x$ (Murray et al., 2012)
emissions are also included.
**2.3.2 Model configurations**
Boundary and initial conditions for WRF employed three alternative meteorological analyses.
They were (1) the National Centers for Environmental Prediction (NCEP)-Final Analysis (FNL)
(https://rda.ucar.edu/datasets/ds083.3/), (2) the fifth generation of European Centre for Medium-
Range Weather Forecasts (ECMWF) atmospheric reanalysis (ERA5) data
(https://rda.ucar.edu/datasets/ds633.0/), and (3) the High-Resolution Rapid Refresh (HRRR)
from NOAA Amazon Web Service (https://registry.opendata.aws/noaa-hrrr-pds). The temporal





resolution for FNL, ERA5, and HRRR is 6-hourly, hourly, and hourly, respectively. The
horizontal resolution for FNL, ERA5, and HRRR is 0.25°, 0.25°, and 3 km, respectively.
WRF has different schemes or options to represent physics and dynamics processes. Three PBL
schemes were used to investigate the effect of different parameterizations of heat, moisture, and
momentum exchange between the surface and PBL on the simulated PBL structure and height.
They are the local closure Mellor-Yamada-Nakanishi-Niino (MYNN) scheme (Nakanishi and
Niino, 2009), the non-local closure Yonsei University (YSU) scheme (Hong et al., 2006), and the
hybrid local-nonlocal Asymmetric Convective Model version 2 (ACM2) scheme (Pleim, 2007).
Details of the PBL schemes are in Sect. 2.3.3. Two microphysics schemes were used: the
Morrison double moment (2M) scheme (Morrison et al., 2009) and the single-moment 6-class
(WSM6) scheme (Hong and Lim, 2006). Other schemes adopted were the Monin-Obukhov
Similarity surface layer, the Noah land surface scheme (Chen and Dudhia, 2001), the Rapid
Radiative Transfer Model (RRTM) longwave and shortwave radiation schemes (Iacono et al.,
2008), and the New Tiedtke cumulus scheme (Zhang et al., 2011; Tiedtke, 1989).
To select the best model configuration to represent meteorology during the 2021 TRACER-AQ
campaign, we designed eight model experiments with different physics options, boundary
meteorology, data assimilation, and reinitializing option, as listed in Table S2. First, [Base] is the
baseline configuration: MYNN for PBL, 2M for microphysics, NCEP FNL for boundary
conditions, no nudging for assimilation, and no reinitialization. Second, [YSU] and [ACM2]
experiments used the YSU and ACM2 PBL schemes, respectively, while keeping other options
the same as [Base]. Differences between [Base], [YSU], and [ACM2] show the effects of
different PBL parameterizations. Third, the [WSM6] experiment differs from [Base] by
replacing the 2M microphysics scheme with WSM6. Differences between [Base] and [WSM6]
show the effects of different microphysics schemes. Next, [ERA5] and [HRRR] were designed to
show the effects of different meteorological initial and boundary conditions on the WRF
performance by using ERA5 and HRRR instead of NCEP FNL, respectively. We examined the
effects of data assimilation options in [Nudged]. [Nudged] adopted observation nudging and
surface analysis nudging to assimilate both onshore and offshore measurements from multiple
platforms, including the TCEQ CAMS, boats, and the NCEP surface and upper air measurements
into WRF meteorology (see Text S2 for details). Differences between [Base] and [Nudged] show
the effects of assimilation. Last, [Reinit] used daily reinitialization where the simulation was
broken into many 30-hour segments with the first 6 hours of each segment (18:00-23:00
Coordinated Universal Time (UTC) of a previous day) as spin-up and the subsequent 24 hours
(0:00-23:00 UTC of the following day) used for analysis (Yahya et al., 2015; Otte et al., 2008).
Differences between [Base] and [Reinit] show the effects of a free-running option versus model
reinitialization.
Among the above simulations, we chose four simulations (Table 1), including the three
simulations with different PBL schemes and the best simulation [HRRR] determined by
campaign-wide statistics (see Text S3 for details) in the analysis below. The surface layer, land
surface, longwave and shortwave radiation, and Tiedtke cumulus schemes remain unchanged in
all simulations.





Table 1. List of simulations used in this study.

| Simulations | Meteorology for Boundary Condition | PBL scheme |
|---|---|---|
| [Base] | NCEP FNL | MYNN |
| [YSU] | NCEP FNL | YSU |
| [ACM2] | NCEP FNL | ACM2 |
| [HRRR] | HRRR | MYNN |

**2.3.3 Determination of PBL height in different schemes**
The heights of the PBL are determined differently among different PBL schemes in the WRF
model. The intra-scheme differences can originate from (1) the vertical profile of thermodynamic
quantities simulated with different assumptions of the vertical exchange of heat, moisture, and
momentum and (2) the diagnosis of the PBL height from these thermodynamic quantities. The
PBL heights determined by different schemes can differ by 20-30% (Hu et al., 2010; Xie et al.,
11 2013).
First, the common parameterizations of vertical exchange include local and non-local closure
schemes. Local closure schemes estimate the turbulent fluxes at each point in model grids from
the mean atmospheric variables and their gradients at that point. In contrast, non-local closure
schemes include the nonlocal upward transport by buoyant plumes, representing large-scale
motions. Among the three PBL schemes used in this study, the MNYY scheme is local, the YSU
is nonlocal, and the ACM2 is hybrid local-nonlocal.
Second, the bulk Richardson number (BRN) and the turbulent kinetic energy (TKE) methods are
the two common methods to determine PBL height. The BRN method diagnoses PBL height
thermodynamically by potential temperature with wind speeds and is adopted in the YSU and the
ACM2 schemes. The PBL heights under this condition are defined as the height of the model
layer where the bulk Richardson number reaches a critical value. The two schemes have two
major differences. The YSU scheme calculates the bulk Richardson number starting from the
surface while the ACM2 scheme calculates it above the neutral buoyancy level (Hu et al., 2010;
Hong et al., 2006; Pleim, 2007). The critical value is 0.25 for stable conditions and 0 for unstable
conditions in the YSU scheme and it is 0.25 for both stable and unstable conditions in the ACM2
scheme (Xie et al., 2013). Meanwhile, the TKE method diagnoses PBL height by horizontal and
vertical winds and is adopted in the MYJ scheme (not used in this study). The PBL height under
this condition is diagnosed when the TKE decreases to a minimum of 0.1 $m^2\ s^{-2}$. A hybrid
definition that combines the BRN and the TKE methods is implemented in the MYNN scheme.
The hybrid method weights the TKE method more during stable conditions when the BRN-based
PBL height is below ~0.5 km, while it weights the TKE-based definition negligible when the
BRN-based PBL height is above ~1 km.
Previous studies have demonstrated that the mentioned schemes outperform each other under
different conditions across regions, evaluated with various metrics (Hu et al., 2010; Xie et al.,
2012; Xie et al., 2013). No conclusion is reached as to which scheme is universally the best. No
systematic higher or lower PBL height is expected from one scheme relative to one another.





**3. Lower tropospheric layering for ozone-episode and non-ozone-episode days**

The geopotential heights at 850 hPa in Figure 2 show different synoptic conditions are seen
between ozone-episode and non-episode days in September 2021. The non-episode days
experienced clean southerlies from the Gulf of Mexico (Fig. 2a), while the ozone episodes of
September 6-11 and 23-26 both happened after a cold frontal passage with a low pressure sitting
in the northeast US and a high pressure located in eastern Texas (Fig. 2b, 2c). This synoptic
structure puts the Houston region under northerly wind conditions, which bring colder and more
polluted continental air to the region, leading to relatively lower temperature (Fig. 3b) and
relative humidity (Fig. 3d) than non-episode days (Fig. 3a; Fig. 3c).

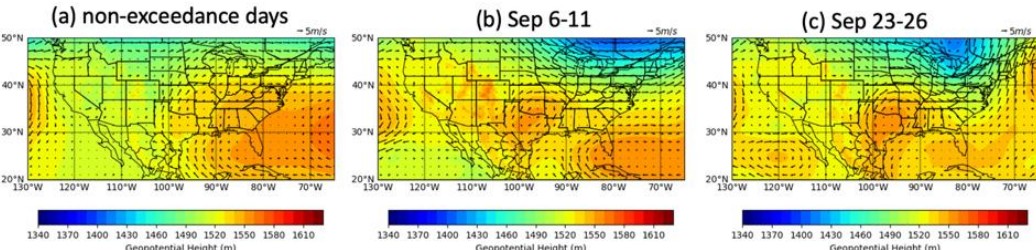

Figure 2. Synoptic conditions denoted by geopotential height at 850 hPa and the associated
winds for (a) the non-episode days and the two ozone episodes of (b) September 6-11 and (c)
September 23-26, 2021.

Apart from differences in meteorological variables, synoptic high-pressure centers during ozone
episodes tend to create a clear, calm condition with light horizontal winds at night when the RL
and the multiple layer structure of the lower troposphere (incl. an SBL, an RL, and a CI layer)
are prone to form, while the RL structure tends to be disrupted due to shear effects under
meteorological conditions during non-episode days (Stull et al., 1988; Yi et al., 2001). We find
mixed layer heights derived from the ceilometer at La Porte during non-episode days (Fig. 3i)
and ozone-episode (Fig. 3j) are similar during the daytime, while the nocturnal mixed layer
heights (*e.g.* 0:00–10:00 CDT) are greater on ozone-episode days than on non-episode days.
Such differences can also be seen from the direct measurements of the ceilometer, i.e.
atmospheric backscatter profiles, in Figures 3k and 3l. During ozone episodes, the high-pressure
center creates favorable meteorological condition for multiple nocturnal layers to form. Among
these, the RL contains much of the aerosol remnant left by the daytime CBL and is therefore
detected by the ceilometer during ozone episodes (Fig. 3l). In contrast, no such multiple layers
form under meteorological conditions on non-episode days. Much of the aerosol remnant above
the SBL is dissipated with the disruption of RL by wind shear such that the SBL contains more
aerosol than above. Therefore, the ceilometer detects the SBL on non-episode days (Fig. 3k). In
this study, the mixed layer heights derived from the ceilometer represent the RL during ozone
episodes but the SBL during non-episode days.

Mixed layer height is often a good proxy for the heights of different lower tropospheric layers
determined thermodynamically in models (Scarino et al., 2014; Kuik et al., 2016; Haman et al.,
2014). We refer to the standard mixed layer retrievals, that is the CBL during the daytime, the
SBL at night during non-episode days, and the RL at night during ozone episodes, respectively



as observed CBL, SBL or RL hereafter in a manner consistent with the modeled equivalents. The
next section evaluates the observed and the modeled PBL heights.
**4. PBL height evaluation**
In this section, we evaluate the modeled PBL height with two types of independent field
measurements. The ground-based ceilometer at the La Porte site is used to evaluate the diurnal
variation, given its continuous measurements throughout the day. Meanwhile, the HSRL-2
instrument acquired data over much of the urban Houston region and adjacent waters and is thus
used to evaluate spatial and temporal (daytime) variations of the PBL.
**4.1 Evaluation with ceilometer**
Diurnal variations of the PBL heights averaged during non-episode periods and ozone episodes
are separately evaluated in Figure 3. The observations represent the daytime CBL on both types
of days. At night, the observations represent the SBL on non-episode days but the RL on ozone-
episode days. The modeled equivalents are needed to yield equal comparisons between the
models and observations. The model diagnoses the CBL height as a standard output for the PBL
height during the daytime. However, the model only diagnoses the SBL as the standard output
for nighttime PBL rather than other nocturnal layers such as RL (Fig. 3j). Therefore, the modeled
RL needs to be extracted for a valid comparison with the observed RL during ozone episodes.

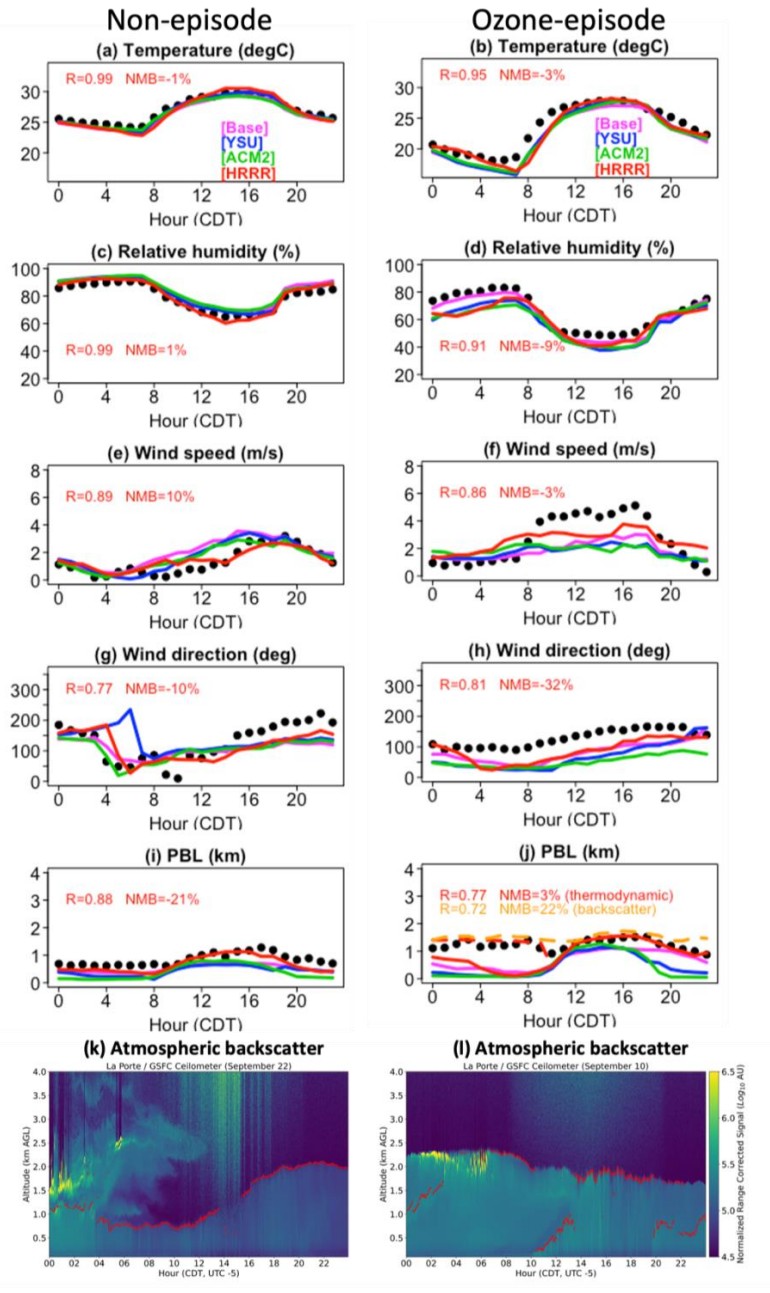

Figure 3. Diurnal variation in observed versus modeled surface meteorology and PBL height
averaged over different days during non-episode periods (left) and ozone episodes (right) in
September 2021. For the first five rows, black dots are NASA GSFC observations at the La Porte
site while lines are equivalent data simulated by the WRF-GC model. Different colors denote
different model configurations. In (j), dashed lines represent residual layers identified by aerosol
backscatter and potential temperature from the [HRRR] configuration. The last row shows the



ceilometer-measured atmospheric backscatter profiles overlaid with mixed layer heights of two
typical days; non-episode day September 22 (k) and ozone-episode day September 10 (l).
Before extracting RL characteristics in the model, we first selected one simulation with the best
daytime performance among the four simulations and examine its nighttime performance. On
non-episode days (Fig. 3i), [Base], [YSU], [ACM2], and [HRRR] respectively show the diurnal
mean and standard deviation of the PBL height of 0.52±0.14 km, 0.43±0.17 km, 0.39±0.27 km,
and 0.66±0.28 km in comparison with the observed value of 0.83±0.22 km. On ozone-episode
days (Fig.3j), the model simulations show the CBL height variation of 0.96±0.18 km ([Base]),
0.60±0.37 km ([YSU]), 0.50±0.5 km ([ACM2]), and 1.25±0.29 km ([HRRR]) in comparison
with the observed value of 1.26±0.24 km during the afternoon and evening hours (15:00-23:00
CDT). During the same period, the model simulations show the PBL decay rates of 53 m h$^{-1}$
([Base]), 102 m h$^{-1}$ ([YSU]), 135 m h$^{-1}$ ([ACM2]), 59 m h$^{-1}$ ([HRRR]) in comparison with the
observed 60 m h$^{-1}$. All model simulations generally underestimate the PBL: 180–450 m
throughout the day on non-episode days and 10–760 m during the daytime on ozone-episode
days. The model underestimations are relative to the observed mixed layer height. The actual
PBL biases in the model can be larger or smaller than those underestimations depending on the
relationship between the backscatter-defined mixed layer and the thermodynamically defined
CBL. Among the four simulations, the [HRRR] best captures the observed mean height and
decay rate during the daytime and is thus examined further for the nighttime hours.
Second, the simulated aerosol backscatter (Fig. 4b, 5b) and potential lapse rates (Fig. 4c, 5c) of
the [HRRR] simulation are used to extract the modeled RL heights. The modeled aerosol
backscatter shows the volume of the atmosphere in which aerosol species are mixed and
dispersed. Substantially stronger backscatter signals are found within the first ~2 km than the
free troposphere at 3-4 km aloft with background backscatter of 0.01-0.02 km$^{-1}$ sr$^{-1}$. Therefore,
we take the sharpest vertical gradient in the backscatter signal (i.e. the largest first derivative of
backscatter) to estimate the modeled mixed layer height. The extracted layers have daytime
variations of 1.58±0.13 km and nighttime variations of 1.50±0.06 km during ozone episodes. The
modeled aerosol backscatter in Figures 4b and 5b is not equivalent to the ceilometer-measured
atmospheric backscatter, which includes both aerosol and molecular backscatter signals in
Figures 4a and 5a. The modeled aerosol backscatter is presented here instead of the total
atmospheric backscatter because the latter is not diagnosed by the model. The modeled aerosol
backscatter is the closest product from the model to denote the modeled mixed layer heights.
Potential lapse rate or potential temperature gradient $\left(\frac{d\theta}{dz}\right)$, defined thermodynamically as the
changes of potential temperature ($\theta$) with height ($z$), is commonly used to distinguish
atmospheric layers according to their instability. Figures 4c and 5c show that the modeled
nocturnal PBL consists of a stable SBL, a neutrally stratified RL, and a CI layer during most
ozone-episode days. The modeled RL top is identified from where the RL (little or low
temperature increases at 0-3 °C km$^{-1}$) shifts to the CI layer (drastic temperature increases at 8-
14 °C km$^{-1}$). Therefore, it can be identified by the sharpest gradient in the potential lapse rate,
which is 6.6 °C km$^{-1}$ on average. The modeled RL top identified here has a variation of
1.39±0.03 km during ozone episodes, slightly lower than the 1.50±0.06 km identified by
backscatter.



The observed RL validates the backscatter-identified and thermodynamically-identified layers
from the model in Figure 3j. Model results have a slightly better agreement with the ceilometer
defined MLH for the thermodynamically-identified layer, with a correlation coefficient (R) of
0.77 and normalized mean bias (NMB) of 3%, than for the backscatter-identified layer, with
R=0.72 and NMB=22%, during ozone episodes.

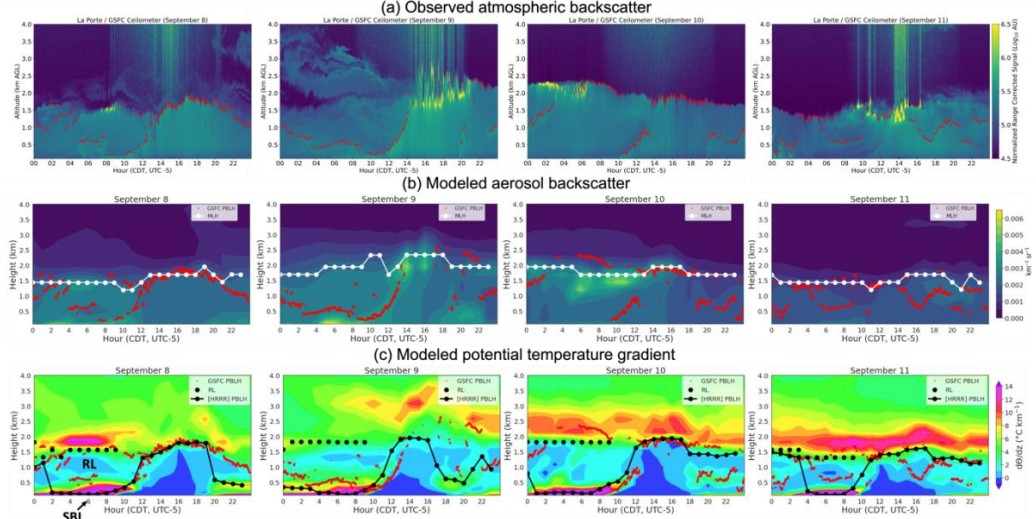

Figure 4. Observed and modeled heights of lower tropospheric layers at the La Porte site during
September 8-11. The contours show (a) ceilometer-observed attenuated atmospheric backscatter
signal produced by aerosols and molecules combined at 1064 nm, (b) modeled unattenuated
backscatter of aerosols alone at 1000 nm, and (c) modeled potential temperature gradient. Red
dots are ceilometer-observed mixed layer. White and black lines are backscatter-defined and
thermodynamically-defined mixed layers from the [HRRR] model simulation.

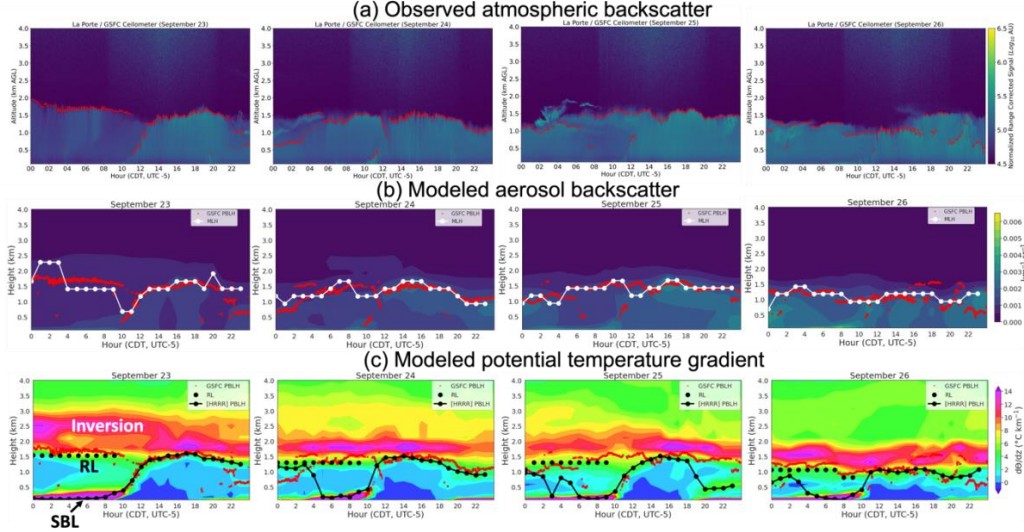

Figure 5. Same as Figure 4 but for September 23-26.





## 4.2 Evaluation with HSRL-2

This section evaluates spatial and temporal (daytime) variations of the modeled PBL heights with the HSRL-2 mixed layer heights. The HSRL-2 collected measurements over the Houston region and the adjacent Galveston Bay typically three times per day for ten days in September 2021. As stated in sections above, the mixed layer can represent the PBL under different conditions and we refer to the observed mixed layer heights as observed PBL hereafter in a manner consistent with the modeled equivalents.

The four model simulations underestimate the observed PBL heights under most conditions. During ozone episodes, the urban Houston region observes the PBL heights of 0.63±0.25 km in the morning (8:00-10:00 CDT), 1.27±0.38 km at noon (11:00-13:00 CDT), and 1.69±0.23 km in the afternoon (14:00-16:00 CDT). The observed heights are consistently lower over water with values of 0.44±0.34 km, 1.14±0.45 km, and 1.43±0.24 km for the three time periods, respectively. Compared to observations, the four model simulations underestimate the heights both over land (by 0.24-0.39 km in the morning, 0.02-0.25 km at noon, and 0.06-0.37 km in the afternoon) and over water (by 0.02-0.25 km in the morning, 0.23-0.59 km at noon and 0.30-0.60 km in the afternoon). During non-episode days, the observed PBL height over land is 0.78±0.14 km in the morning and 1.07±0.24 km at noon. The morning heights are underestimated by 0.10-0.34 km in the four model simulations, while the noon heights can be either underestimated by 0.25-0.37 km or overestimated by 0.05-0.23 km. In contrast to land, the observed height over water is consistently lower with values of 0.57±0.28 km in the morning and 0.65±0.34 km at noon; these are underestimated in the model by 0.03-0.28 km and 0.11-0.21 km, respectively. Among the four simulations, the [HRRR] is the best to reproduce observed values with the minimum model-observation differences under most conditions (e.g. different times and regions), as shown in Table 2.

The observed land-water differences in PBL heights are larger in the afternoon than in the morning during both ozone-episode and non-episode days. The four model simulations capture the land-water differences better on non-episode days than ozone-episode days. During ozone episodes, the observed mean land-water difference (land minus water) is 0.19 km while the model predicts smaller differences of –0.05~0.06 km in the morning; this is because the model shows consistent larger underestimations over land than water. During noon and afternoon hours, the observed mean land-water differences of 0.13 km and 0.26 km are predicted to be higher in the model with values of 0.32-0.52 km and 0.44-0.56 km, respectively; this is because the model shows consistent smaller underestimations over land than water during this period. During non-episode days, the observed land-water gradients of 0.21 km (morning) and 0.42 km (noon) are simulated to be 0.14-0.22 km and 0.36-0.76 km, respectively. The [ACM2] and the [HRRR] slightly outperform the other two simulations for land-water differences (Table 2).

One ozone-episode day, September 9, is selected to show the spatial characteristics of different simulations (Fig. 6). The four simulations match the observed mixed layer heights with high spatial correlation at noon (R=0.62-0.77) and in the afternoon (R=0.71-0.82). Among all simulations, the [HRRR] and the [Base] show the best spatial correlations at noon and in the afternoon, respectively. The [HRRR] shows sharp transitions for the different flight legs over urban Houston at noon on September 9 because of drastic changes in the modeled PBL heights at





an hourly interval. The morning mixed layer heights can be difficult to retrieve with the
influences from multiple layers (e.g. SBL and RL), and they can differ substantially from the
thermodynamically-defined PBL. Therefore, we do not expect the model to capture the spatial
patterns of mixed layer heights in the morning.
Table 2. Differences of the HSRL-2 mixed layer height and the WRF-GC thermodynamic PBL
height during ozone-episode days (September 8-10 and 23-26) and non-episode days (September
1 and 3). Land and water are defined by the gray boxes in Figure 6. The bias difference and the
root mean square (RMS) difference are calculated by model simulations minus HSRL-2, with the
unit of kilometers.

| | Simulations | Differences | Morning (8:00-10:00 CDT) | | Noon (11:00-13:00 CDT) | | Afternoon (14:00-16:00 CDT) | |
|---|---|---|---|---|---|---|---|---|
| | | | Land | Water | Land | Water | Land | Water |
| Ozone-episode | [Base] | Bias | -0.242 | -0.086 | -0.240 | -0.497 | -0.372 | -0.578 |
| | | RMS | 0.381 | 0.318 | 0.436 | 0.668 | 0.464 | 0.696 |
| | [YSU] | Bias | -0.392 | -0.250 | -0.194 | -0.589 | -0.301 | -0.610 |
| | | RMS | 0.488 | 0.409 | 0.406 | 0.785 | 0.409 | 0.807 |
| | [ACM2] | Bias | -0.294 | -0.167 | -0.076 | -0.471 | -0.278 | -0.457 |
| | | RMS | 0.430 | 0.378 | 0.376 | 0.683 | 0.476 | 0.665 |
| | [HRRR] | Bias | -0.262 | -0.026 | -0.040 | -0.232 | -0.068 | -0.303 |
| | | RMS | 0.384 | 0.312 | 0.289 | 0.462 | 0.223 | 0.455 |
| | Simulations | Differences | Morning | | Noon | | Afternoon | |
| | | | Land | Water | Land | Water | Land | Water |
| Non-episode | [Base] | Bias | -0.211 | -0.218 | -0.243 | -0.276 | | |
| | | RMS | 0.309 | 0.353 | 0.433 | 0.440 | | |
| | [YSU] | Bias | -0.348 | -0.282 | -0.363 | -0.304 | | |
| | | RMS | 0.434 | 0.397 | 0.506 | 0.471 | | |
| | [ACM2] | Bias | -0.236 | -0.236 | 0.050 | -0.261 | | |
| | | RMS | 0.371 | 0.377 | 0.460 | 0.440 | | |
| | [HRRR] | Bias | -0.100 | -0.029 | 0.237 | -0.107 | | |
| | | RMS | 0.243 | 0.301 | 0.377 | 0.364 | | |



Figure 6. Spatial variabilities of the PBL heights (in meters) from the HSRL-2 and different
WRF-GC simulations (a) in the morning (8:00-10:00 CDT), (b) at noon (11:00-13:00 CDT), and
(c) in the afternoon (14:00-16:00 CDT) of September 9, 2021.





**5. Ozone vertical mixing and distribution**

Boundary layer mixing can bring air aloft towards the surface and vice versa, leading to uneven vertical distribution of ozone which accordingly affects surface ozone concentrations. This section uses independent field measurements at La Porte (incl. TROPOZ, HSRL-2, ozonesondes, a model 49i ozone analyzer, and a CAMS site named La Porte Sylvan Beach) to validate the modeled vertical ozone profiles at three layers, including the lower free troposphere (2-3 km aloft), the boundary layer (0.5-1 km aloft), and the ground level (<50 m). Since the [HRRR] simulation best represents the PBL variations in Section 4, it is used to investigate vertical ozone profiles in this section.

**5.1 Free tropospheric ozone entrainment**

The strength of the CI layer regulates the gas exchange between the FT and the PBL. Strong convection can penetrate a weak CI layer and entrain FT air into the PBL (i.e. entrainment), while a strong CI layer acts as a lid to restrict gas exchange between the PBL and the FT. The potential temperature differences between the top and bottom of the CI layer are often used to indicate the strength of the CI layer and the extent of entrainment processes (Kaser et al., 2017; Morris et al., 2010; Rappenglück et al., 2008). We first identified the modeled CI layers at 1.5–3 km aloft during ozone episodes (Fig. 4c and Fig. 5c), and then calculate the temperature differences in the model between the top and bottom of the CI layers in each day. The corresponding daily inversion strength is 2.3 °C, 2.8 °C, 6.8 °C and 6.4 °C during September 8-11 and 13.6 °C, 7.5 °C, 7.8 °C, and 8.4 °C during September 23-26, respectively. Among these days, September 8 and 9 experienced the weakest inversions. To examine if the modeled inversion strength is representative of the observations, we evaluate the modeled potential temperature profiles with ozonesonde measurements in Fig. 7a. Results show that the model simulates the vertical profiles of potential temperature well across different days with high correlation (R=0.99) and low biases (MB= -0.64 °C ~ -0.17 °C).

Combining the inversion strengths (Fig. 4c; Fig. 5c) and the vertical ozone distributions from the TROPOZ lidar (Fig. 8) helps to identify potential entrainment of the FT air into the underlying PBL on September 8 and 9 at the La Porte site. On September 8, strong convection associated with a rapid CBL growth penetrates the thin and weak inversion at 2 km aloft at around noon (Fig. 4c) and allows the ozone-rich air above to entrain into the CBL, adding to afternoon ozone buildup (Fig. 8a). Similarly, there is no CI layer present overnight from 20 CDT on September 8 to 10 CDT on September 9 (Fig. 4c) and thus long-lasting ozone entrainment into the RL (Fig. 8a). Conversely, a strong and thick inversion at 1.5-3 km decouples the FT and the underlying PBL during September 23-24 (Fig. 5c) and the ozone layer remains aloft at 2-3 km (Fig. 8d). The inversion strength presented here is one way to approach the potential entrainment, follow-up studies can probe into the detailed dynamics. It is also noteworthy that the presented vertical distribution of ozone is also largely shaped by local ozone production in the boundary layer. Since this study is focused on the vertical ozone distribution impacted by mixing between lower free tropospheric layers, the vertical ozone distribution impacted by chemistry and differentiating between the contributions from dynamics and chemistry are outside the scope of this analysis.

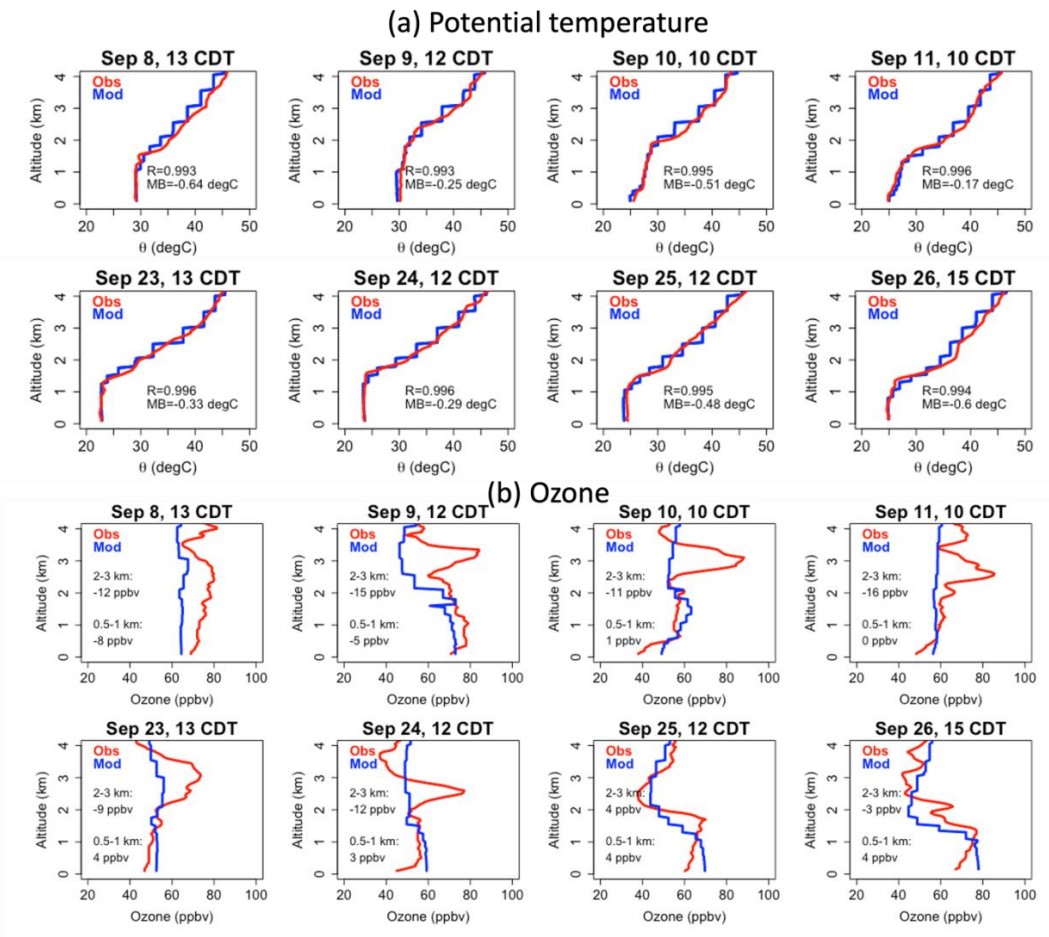

Figure 7. Vertical profiles of (a) potential temperature and (b) ozone from ozonesonde
measurements and the WRF-GC [HRRR] simulation at La Porte during September 8-11 and
September 23-26.



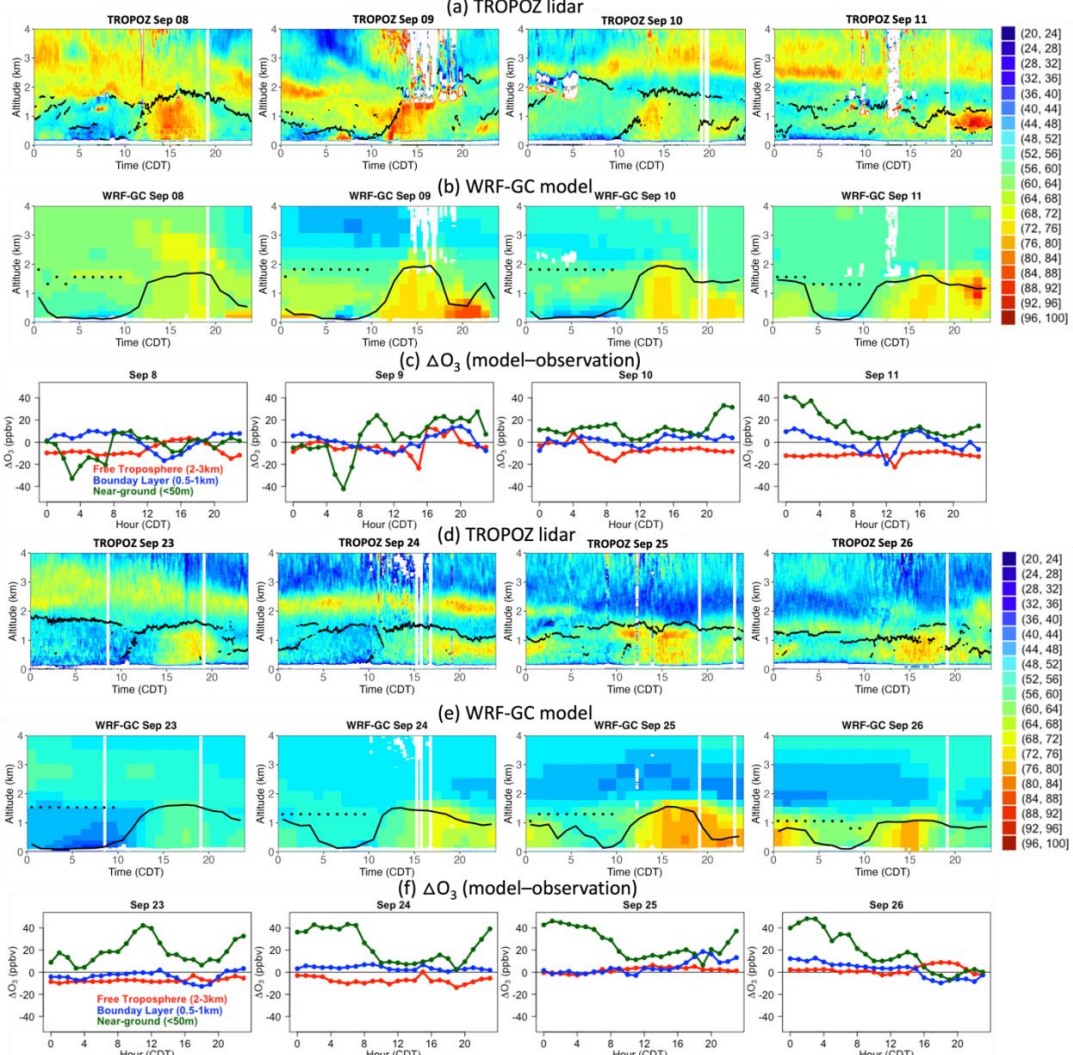

Figure 8. Time series of the vertical ozone profile from the TROPOZ ozone lidar (a, d) and the
WRF-GC [HRRR] simulation (b, e) at La Porte. Observed and modeled boundary layer heights
are inserted, respectively. Dots represent the modeled residual layer identified in this study. Line
plots (c, f) show ozone differences (model minus observation) at the free troposphere (2-3 km)
and the boundary layer (0.5-1 km) from the TROPOZ as well as the near-ground (< 50m) from
the model 49i ozone analyzer.



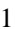

Figure 9. Vertical ozone profiles from (a, d) the HSRL-2 and (b, e) the WRF-GC [HRRR]
simulation. The profiles are taken from a flight track (Fig.1) over urban Houston and Galveston
Bay at around 11:00-13:00 CDT each day. Line plots (c, f) show ozone differences (model minus
observation) at the free troposphere (2-3km) and the boundary layer (0.5-1km).



**5.2 Evaluation of ozone vertical distribution**

Multiple field measurements at La Porte are used to evaluate the modeled vertical ozone distribution at the free troposphere (2-3 km), the boundary layer (0.5-1 km), and the near-ground level (< 50m). According to data availabilities at different levels, the free troposphere and boundary layer are evaluated by the TROPOZ, the HSRL-2, and ozonesondes (Table 3), while the ground level is evaluated by a model 49i ozone analyzer and a CAMS site named La Porte Sylvan Beach (Table 4). To cross compare among multiple measurements, we present the model-observation differences at a common site (La Porte) during a common time slot (11:00-13:00 CDT). Larger ozone differences are found at the near-ground level than for the boundary layer and lower free troposphere (Table 3; Table 4).

As shown in Table 3, the model underestimates the layer of enhanced ozone at 2-3 km aloft in the free troposphere by 9%-21% (TROPOZ), 15%-22% (HSRL-2), and 14%-22% (ozonesondes) at La Porte at 11:00-13:00 CDT during September 8-11 and September 23-24. Unlike most of the campaign's ozone exceedance days, September 25 and 26 do not have an enhanced ozone layer at 2-3 km aloft but have a lower ozone layer relative to the background tropospheric values instead; this low ozone layer is overestimated by 9%-12% on September 25 but underestimated by 3-12% on September 26. Meanwhile, the model underestimates the boundary layer ozone at 0.5-1 km aloft by 6%-10% (TROPOZ), 6%-12% (HSRL-2), and 1%-11% (ozonesondes) during the first ozone episode of September 8-11, but overestimate it by 0%-6% (TROPOZ), 3%-5% (HSRL-2), and 5%-7% (ozonesondes) during the second episode of September 23-26. Above model-observation differences are based on the common site (La Porte) and common time (11:00-13:00 CDT) among different measurements, the temporal (Figures 8c and 8f) and spatial (Figures 9c and 9f) variations of these differences are shown in Figures 8 (TROPOZ) and 9 (HSRL-2).

While free tropospheric and boundary layer ozone are important components of the vertical ozone distribution due to their thickness, the thin layer of near-ground ozone affects human and vegetation health the most and thus receives more attention. In Table 4, the model overestimates near-ground ozone by 6%-24% (model 49i ozone analyzer) and 8%-39% (CAMS La Porte Sylvan Beach) at La Porte at 11:00-13:00 CDT during the two ozone episodes. Figures 8c and 8f show the temporal variations of model-observation differences from the model 49i ozone analyzer. Most near-ground ozone differences occur at night, consistent with the known problem of overestimating nighttime ozone common to many photochemical models (Schnell et al., 2015; Travis et al., 2016; Jaffe et al.,2018). The WRF-GC model adopts a chemical module from GEOS-Chem. Thus, the two share the difficulties replicating nighttime ozone due to reasons such as the insufficient representation of the stratification of multiple nocturnal atmospheric layers, uncertainties in gas exchanges between the residual layer and the underlying surface layer, and difficulties in simulating the timing of changes in PBL dynamics (Travis and Jacob, 2019).



Table 3. Absolute (abs.) and percentage (pct.) ozone differences between field measurements and
the model at free troposphere and boundary layer at La Porte.

| | | TROPOZ (11-13 CDT) | | HSRL-2 (11-13 CDT) | | Ozonesonde (10-15 CDT) | |
|---|---|---|---|---|---|---|---|
| | | abs. (ppbv) | pct. | abs. (ppbv) | pct. | abs. (ppbv) | pct. |
| Free Troposphere (2-3km aloft) | 8-Sep | -7 | -9% | -12 | -15% | -12 | -16% |
| | 9-Sep | -8 | -13% | -13 | -20% | -15 | -22% |
| | 10-Sep | -8 | -13% | -14 | -21% | -11 | -18% |
| | 11-Sep | -16 | -21% | | | -16 | -21% |
| | 23-Sep | -8 | -13% | -11 | -17% | -9 | -14% |
| | 24-Sep | -9 | -15% | -14 | -22% | -12 | -20% |
| | 25-Sep | 5 | 12% | 6 | 15% | 4 | 9% |
| | 26-Sep | -2 | -3% | -6 | -12% | -3 | -6% |
| Boundary Layer (0.5-1km aloft) | 8-Sep | -5 | -7% | -8 | -11% | -8 | -11% |
| | 9-Sep | -8 | -10% | -5 | -6% | -5 | -7% |
| | 10-Sep | -4 | -6% | -9 | -12% | 1 | 2% |
| | 11-Sep | -5 | -7% | | | -0.4 | -1% |
| | 23-Sep | 0 | 0% | -2 | -3% | 4 | 7% |
| | 24-Sep | 2 | 4% | 2 | 4% | 3 | 5% |
| | 25-Sep | 1 | 2% | 2 | 3% | 4 | 7% |
| | 26-Sep | 4 | 6% | 3 | 5% | 4 | 5% |

Table 4. Absolute (abs.) and percentage (pct.) ozone differences between field measurements and
the model at the near-ground level at La Porte.

| | | Model 49i (11-13 CDT) | | CAMS La Porte Sylvan Beach (11-13 CDT) | |
|---|---|---|---|---|---|
| | | abs. (ppbv) | pct. | abs. (ppbv) | pct. |
| Near-ground (< 50m) | 8-Sep | 4 | 7% | | |
| | 9-Sep | 8 | 12% | | |
| | 10-Sep | 4 | 6% | 16 | 31% |
| | 11-Sep | 6 | 9% | 15 | 29% |
| | 23-Sep | 4 | 9% | 4 | 8% |
| | 24-Sep | 10 | 20% | 15 | 34% |
| | 25-Sep | 13 | 24% | 18 | 33% |
| | 26-Sep | 12 | 21% | 21 | 39% |



**6. Conclusion**
We used ground-based and aircraft observations collected during the TRACER-AQ campaign in
September 2021 to evaluate WRF-GC simulation of the PBL height and ozone in Houston,
including two ozone episodes characterized by MDA8 ozone exceeding 70 ppbv. The combined
suite of ground-based and airborne meteorological and chemical observations are critical in
thoroughly evaluating the spatial and temporal variations of the PBL and vertical ozone
distributions during multi-day ozone episodes, as presented in this work.
The modeled PBL heights are evaluated with mixed layer heights retrieved by the ground-based
ceilometer and the airborne HSRL-2. Compared with the ceilometer, the four model simulations
of [Base], [YSU], [ACM2], and [HRRR] generally underestimate the PBL heights to different
extents: 180–450 m throughout the day on non-episode days and 10–760 m during the daytime
on ozone-episode days. As the best simulation, the [HRRR] captures the diurnal variations
during non-episode days (R=0.88; NMB=-21%). Standard models do not diagnose RL heights,
unlike ceilometers. Therefore, we separately identified the modeled RL following the practices
using aerosol backscatter signals and potential temperature gradients during the ozone episodes.
As a result, the diurnal variation of the thermodynamically-identified layer (R=0.77; NMB=3%)
compares slightly better than that of the backscatter-identified layer (R=0.72; NMB=22%)
during ozone episodes. Meanwhile, when compared with the HSRL-2, the four simulations
underestimates PBL heights by 20-390 m over the urban Houston region and by 20-600 m over
the adjacent Galveston Bay during ozone episodes. On non-episode days, the PBL heights over
urban region are either underestimated by 100-370 m or overestimated by 50-230 m and those
over the Bay are underestimated by 30-210 m by the four simulations. On both ozone-episode
and non-episode days, the observed land-water differences in PBL heights are larger in the
afternoon than in the morning; the model captures such daytime trends. The four model
simulations capture the land-water differences better on non-episode days than ozone-episode
days.
We evaluated the vertical ozone distribution with multiple field measurements, including
TROPOZ, HSRL-2, ozonesonde, a model 49i ozone analyzer, and a CAMS site named La Porte
Sylvan Beach. Evaluations were done at three lower tropospheric layers: the free troposphere (2-
3 km aloft), the boundary layer (0.5-1 km aloft), and the ground level (< 50 m aloft). As a result,
the model underestimates the high ozone layer in the free troposphere by 9%-21% (TROPOZ),
15%-22% (HSRL-2), and 14%-22% (ozonesondes) on most ozone-episode days. The boundary
layer ozone is underestimated by 6%-10% (TROPOZ), 6%-12% (HSRL-2), and 1%-11%
(ozonesondes) during September 8-11, but overestimated by 0%-6% (TROPOZ), 3%-5%
(HSRL-2), and 5%-7% (ozonesondes) during September 23-26. Meanwhile, the model
overestimates near-ground ozone by 6%-24% (model 49i ozone analyzer) and 8%-39% (CAMS
La Porte Sylvan Beach) during the two ozone episodes.
Based on these evaluations, we summarized model limitations that prevent more accurate
simulation of PBL heights and the vertical ozone distribution during TRACER-AQ. The first
limitation is the single-layer PBL representation. The WRF model only diagnoses the SBL at
night, despite the model simulating different physical and thermodynamic properties of multiple
nocturnal layers above the SBL. For example, the RL is not identified by the model as a standard
diagnosis; this prevents the direct comparison of the model outputs with the observed RL at



night. The second limitation is the underestimation of the layer of enhanced ozone 2-3 km aloft
in the free troposphere that was often present on ozone-episode days during the campaign. Given
its height of 2-3 km and a lifetime of around a week, the layer of enhanced ozone was likely
transported into Houston by synoptic flows of cold fronts from the north. The
underrepresentation of the synoptic layer of enhanced ozone affects model representations across
regions horizontally and atmospheric layers vertically, making it particularly important to model
vertical ozone distributions and the effects of entrainment accurately.
Our findings of the model limitations have implications for the predictivity of ozone's vertical
mixing and distribution across different modeling systems. For example, WRF is widely used in
various meteorology-chemistry coupling systems with different treatments of boundary layer
mixing. In WRF-Chem, boundary layer mixing in the chemistry part uses a mixing coefficient
originating in WRF such that the boundary layer mixing calculations in the meteorology and
chemistry parts share the same set of coefficients. In WRF-GC, the chemistry part from GEOS-
Chem only takes the PBL height from WRF as the maximum height for boundary layer mixing
and conducts independent calculations of boundary layer mixing from WRF. Unlike online
coupled WRF-Chem and WRF-GC, WRF is offline coupled to CAMx in the WRF-CAMx
system, and the boundary layer mixing in the chemistry part of CAMx is subject to WRF output
frequency instead of the native transport time step in WRF. Thus, it is essential to understand
how the model limitation of a single-layer PBL representation affects boundary layer mixing in
chemical simulations among different meteorology-chemistry coupling systems. Follow-up
studies to this work will address these aspects with detailed model intercomparisons.
**Code availability.** WRF-GC is a free and open source model (http://wrf.geos-chem.org; last
access: 29 May 2023) (Lin et al., 2019; Feng et al., 2021). The two parent models, WRF and
GEOS-Chem, are also open source and can be obtained from their developers at
https://github.com/wrf-model/WRF (last access: 29 May 2023) and http://www.geos-chem.org
(last access: 29 May 2023), respectively. The version of WRF-GC (v2.0) described in this paper
couples WRF v3.9.1.1 and GEOS-Chem v12.7.2 and is archived in Zenodo at
https://doi.org/10.5281/zenodo.4395258 (last access: 29 May 2023).
**Data availability.** All observation datasets, model configuration files, model boundary
conditions, model input files, and scripts used in this paper are archived in Zenodo at
https://doi.org/10.5281/zenodo.7983449 (last access: 29 May 2023).

**Author contributions.** XL and YW conceived the research idea. XL wrote the initial draft of the
paper and performed the analyses and model simulations. JF, TG, and SA provided the shipborne
data. JS, MR, and LT provided the TROPOZ and ceilometer data. GG and TB provided the
LMOL data. PW and JS provided the ozonesonde data. JH, TS, AJS, and MF provided the
HSRL-2 data. All authors contributed to the interpretation of the results and the preparation of
the paper.

**Competing interests.** The contact author has declared that none of the authors has any
competing interests.



**Acknowledgements.** The authors acknowledge TCEQ for providing the hourly wind,
temperature, relative humidity, and MDA8 ozone data, and NASA Langley Atmospheric Science
Data Center for providing the TRACER-AQ data archive. We thank Richard Ferrare for helpful
suggestions on this paper.
**Financial support.** This research was supported by the Texas Commission on Environmental
Quality (TCEQ) (Grant No. 582-22-31544-019) and by a grant from the Texas Air Quality
Research Program (AQRP) (22-008) at The University of Texas at Austin through the Texas
Emission Reduction Program (TERP) and the TCEQ. The findings, opinions, and conclusions
are the work of the author(s) and do not necessarily represent the findings, opinions, or
conclusions of the AQRP or the TCEQ.

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
