# Peer review of "Evaluating WRF-GC v2.0 predictions of boundary layer height and vertical 1"

_EGUsphere, 2023_

## Referee Comment (RC1)

**Review for manuscript "Evaluating WRF-GC v2.0 predictions of boundary layer and vertical ozone profiles during the 2021 TRACER-AQ campaign in Houston, Texas" by X. Liu et al.**

The manuscript presents a thorough comparison of WRF-GC PBL heights and ozone profile predictions with respect to observations obtained with different remote sensing and in situ instrumentation. The study identified some model limitations which have relevant implications for further modeling, especially for applications related e.g. to human or vegetation health. Therefore, I clearly agree that this study fits within the scientific innovation, quality and the scope of Geoscientific Model Development.

The manuscript presents high scientific level, and significant results. It is mostly well written and well structured, the objective is clear and the approach is technically well justified and validated. The abstract is accurate and concise, the introduction properly presents the topic background, previous works on the subject are properly cited and the new points are clearly indicated. To my view, the description of the observations (section 2.1) is less properly addressed and organized and should be revised by the authors. Description of the used model configurations (section 23) is very clear and the results sections (4 and 5) are well structured and present the relevant information.

I propose that this article is accepted for publication, after improving some aspects that, to my view, will make the work more robust:

-Title: it would be better to specify "boundary layer height", since this is the BL parameter that authors are evaluating.

-Pag. 2 Line 38 – Pag. 3 Line 3: they are actually more types of measured variables used to calculate PBL height. For example, please refer to the recent review of Kotthaus et al. (https://doi.org/10.5194/amt-16-433-2023) and references therein, where different systems and atmospheric variables are discussed, e.g. thermodynamical variables (with MWR, IRS, Raman lidars, RASS) and aerosol profiles (with aerosol lidars, ceilometers), but also gases (e.g. with DIAL), wind and turbulence (with DWL, RWP, Sodar). Please update the information.

-Throughout the whole text, authors use "PBL" (e.g. Page 3 Line 39, Page 4 Line 3, Page 12 Line 14, Page 12 Line 17, Page 14 Line 7) when the refer specifically to PBL height (PBL could be also evaluated referring to other parameters, not only height). Please use "PBL height" or "PBLH".

-Page 4 Line 10: please specify what "semi-continuous" stands for.

-Page 4 Lines 19-22: why is LMOL and University of Houston site mentioned? The measurements are not used for the presented study, but only included in the supplement. If the measurements are relevant for the study (i.e., for the comparison with the model), they should also be discussed in the main manuscript results sections, and the observations should be better explained in the in 2.1 section. If not, they should be removed from the supplement.

-Page 5 Line 7: why there is a distinction between "aerosols" and "aerosol optical properties"?

-Page 5 Lines 14-16: this information is redundant here.

-Page 5 Line 18: here "boats" is very general and not enough information is given previously to identify what the authors mean. There is only a mention to shipborne platforms in Page 3, but there seemed to be related to a different study. Since the shipborne in situ data are used in this study, please include the description also in this section 2.1.

-Page 7 Line 12: "Other schemes adopted…", do you mean in this paper or in the literature?

-Figure S3: are the time series spatial averages? Please specify in the caption

-Table S3: are the values in column "OBS" the spatial and temporal averages? Please specify.

-Figure 3(i) and (j): a lower y-axis scale maximum, e.g., 2 or 2.5 km would make the variability clearer.

-Page 12 Line 26: "0.01-0.02 $km^{-1}sr^{-1}$" are not the values shown in Figures 4 and 5

Other corrections or typos:

-Page 4 Line 26: "(RCS) are shown"

-Page 9 Line 23: "i.e." instead of "e.g."

-Page 12 Line 5: "examined"

-Figure 8 (a), (b), (d), (e): please specify the units of the color scale

-Page 23 Line 20: "underestimated"

---

## Author Response (AR1)

Reply to Reviewers

We thank the two reviewers for their constructive comments and suggestions, which have further improved the quality of our manuscript considerably. Their comments are reproduced below with our responses in blue. The corresponding edits in the manuscript are highlighted with tracked changes. The line numbers are based on the revised manuscript.

**Reviewer #1:**

The manuscript presents a thorough comparison of WRF-GC PBL heights and ozone profile predictions with respect to observations obtained with different remote sensing and in situ instrumentation. The study identified some model limitations which have relevant implications for further modeling, especially for applications related e.g. to human or vegetation health. Therefore, I clearly agree that this study fits within the scientific innovation, quality and the scope of Geoscientific Model Development.

The manuscript presents high scientific level, and significant results. It is mostly well written and well structured, the objective is clear and the approach is technically well justified and validated. The abstract is accurate and concise, the introduction properly presents the topic background, previous works on the subject are properly cited and the new points are clearly indicated. To my view, the description of the observations (section 2.1) is less properly addressed and organized and should be revised by the authors. Description of the used model configurations (section 23) is very clear and the results sections (4 and 5) are well structured and present the relevant information.

Response: We agree with the reviewer's comment and have altered Section 2.1 to improve its clarity.

I propose that this article is accepted for publication, after improving some aspects that, to my view, will make the work more robust:

-Title: it would be better to specify "boundary layer height", since this is the BL parameter that authors are evaluating.

Response: Revised as suggested.

-Pag. 2 Line 38 – Pag. 3 Line 3: they are actually more types of measured variables used to calculate PBL height. For example, please refer to the recent review of Kotthaus et al. (https://doi.org/10.5194/amt-16-433-2023) and references therein, where different systems and atmospheric variables are discussed, e.g. thermodynamical variables (with MWR, IRS, Raman lidars, RASS) and aerosol profiles (with aerosol lidars, ceilometers), but also gases (e.g. with DIAL), wind and turbulence (with DWL, RWP, Sodar). Please update the information.

Response: Thank you so much for this valuable information! We have now incorporated this comprehensive summary into that paragraph. Corresponding references have been added to the manuscript.

P2 L35: "*The heights of the PBL (incl. CBL and SBL) and other lower tropospheric layers (e.g., RL, CI) are defined mainly by temperature inversions. It is primarily a thermodynamic-based definition*, **but various types of measurements can be used to calculate the height of the PBL. These measurements include (1) thermodynamical quantities, (2) atmospheric aerosol particle characteristics, (3) atmospheric gases, and (4) wind and turbulence (Kotthaus et al., 2023). The first type measures**

*thermodynamic properties (e.g., temperature, water vapor mixing ratio, etc.) with microwave radiometer (MWR), infrared spectrometer (IRS), Raman lidar, radio acoustic sounding system (RASS), etc (Cimini et al., 2013; Wulfmeyer et al., 2010). The second type measures backscatter signals (e.g., attenuated backscatter, particle backscatter, etc.) with aerosol lidars, ceilometers, etc (Caicedo et al., 2017; Knepp et al., 2017; Li et al., 2022). The third type measures the mass or number concentration of gases with differential absorption lidar (DIAL) (Hair et al., 2008). The fourth type measures dynamic and turbulent processes (e.g., horizontal wind speed, variances of the velocity components, turbulent kinetic energy (TKE), eddy dissipation rate, etc.) with Doppler wind lidar (DWL), radar wind profiler (RWP), sodar, etc (Bonin et al., 2016; Bodini et al., 2018; Bonin et al., 2018; Angevine et al., 2003).*"

-Throughout the whole text, authors use "PBL" (e.g. Page 3 Line 39, Page 4 Line 3, Page 12 Line 14, Page 12 Line 17, Page 14 Line 7) when the refer specifically to PBL height (PBL could be also evaluated referring to other parameters, not only height). Please use "PBL height" or "PBLH".

Response: Revised as suggested for the entire manuscript.

-Page 4 Line 10: please specify what "semi-continuous" stands for.

Response: Revised to 'continuous' in P4 L12.

-Page 4 Lines 19-22: why is LMOL and University of Houston site mentioned? The measurements are not used for the presented study, but only included in the supplement. If the measurements are relevant for the study (i.e., for the comparison with the model), they should also be discussed in the main manuscript results sections, and the observations should be better explained in the in 2.1 section. If not, they should be removed from the supplement.

Response: We agree with the reviewer's comment and have removed Text S4 and Figure S6 in the supplement.

-Page 5 Line 7: why there is a distinction between "aerosols" and "aerosol optical properties"?

Response: Revised as suggested in P5 L9.

-Page 5 Lines 14-16: this information is redundant here.

Response: This information is now deleted.

-Page 5 Line 18: here "boats" is very general and not enough information is given previously to identify what the authors mean. There is only a mention to shipborne platforms in Page 3, but there seemed to be related to a different study. Since the shipborne in situ data are used in this study, please include the description also in this section 2.1.

Response: Thank you for bringing this to our attention. We have now incorporated this information on "boats" into Section 2.2 under "identification of ozone episodes" in P5 L18 and Text S1. The detailed description on boat observations is given in Li et al. (2023) which extensively describes and utilizes the boat observations to evaluate the photochemical model. Since the boat observations are not the core focus of this study, we have opted not to provide a full description of the boat observations in Section 2.1. Instead, we refer readers to Li et al. (2023) for further information on boat observations.

P5 L18: "*Ozone exceedance days used in this study were identified by the same criteria used in Li et al. (2023) and Soleimanian et al. (2023), where (1) any onshore site from the TCEQ CAMS network in Houston and Galveston or (2)* **offshore ozone observations by boat operating in Galveston Bay during the field campaign** *registered daily maximum 8-hour average (MDA8) ozone in exceedance of the current NAAQS air quality standard of 70 ppbv (**see Text S1 for details; refer to Li et al. (2023) for full description of the boat observations***).*"

-Page 7 Line 12: "Other schemes adopted...", do you mean in this paper or in the literature?

Response: Revised to be '*Other schemes adopted* **in this paper** *...*' in P7 L10.

-Figure S3: are the time series spatial averages? Please specify in the caption

Response: Yes, spatial averages. Caption updated in P6 L4 in the supplement. Clarification of spatial averages of wind speed and direction is also added in Text S3 in P3 L18 in the supplement.

P6 L4 in the supplement: "*Figure S3. Hourly time series of observation-model differences (i.e., model minus observation) are shown for (a) air temperature, (b) relative humidity, (c) wind speed and (d) wind direction.* **The differences are spatial averages across all CAMS stations and the WRF model equivalents during ozone episodes. Refer to Text S3 for the calculations of spatial averages of wind speed and directions, as well as the differences between observed and modeled wind directions.**"

P3 L18 in the supplement: "**The mean of wind speed and direction is calculated using the vector notation approach, a commonly used method in wind evaluations, as described in Yu et al. (2023). This method treats wind as vectors with their u (eastward) and v (northward) wind components. First, the mean u and v wind components are found by averaging all u and v wind values over a given time period. Then, the resultant vector is determined by taking the square root of the sum of the squares of the mean u and mean v wind components. The magnitude of resultant vector represents the mean wind speed, and the angle of the resultant vector represents the mean wind direction.**

**The difference between observed and modeled wind direction was calculated as below.**

$$\Delta = \begin{cases} M - O, & \text{when } |M - O| \leq 180° \\ (M - O)\left(1 - \dfrac{360}{|M - O|}\right), & \text{when } |M - O| > 180° \end{cases}$$

**where M is the model output, and O is the observation. The correlation between observed and modeled wind direction was determined by a circular correlation coefficient as below.**"

$$R = \frac{\sum_{i=1}^{N} \sin(M_i - \bar{M})\sin(O_i - \bar{O})}{\sqrt{\sum_{i=1}^{N} \sin^2(M_i - \bar{M})}\sqrt{\sum_{i=1}^{N} \sin^2(O_i - \bar{O})}}$$

-Table S3: are the values in column "OBS" the spatial and temporal averages? Please specify.

Response: Yes, spatial and temporal averages. Clarification is now added in the caption of Table S3 in P7 L5 in the supplement.

P7 L5 in the supplement: "**OBS and MOD represent the spatial and temporal averages of observations and model equivalents, respectively.**"

-Figure 3(i) and (j): a lower y-axis scale maximum, e.g., 2 or 2.5 km would make the variability clearer.

Response: Revised as suggested.

-Page 12 Line 26: "0.01-0.02 $km^{-1}sr^{-1}$" are not the values shown in Figures 4 and 5

Response: Revised as suggested.

Other corrections or typos:
-Page 4 Line 26: "(RCS) are shown"

Response: Revised as suggested.

-Page 9 Line 23: "i.e." instead of "e.g."

Response: Revised as suggested.

-Page 12 Line 5: "examined"

Response: Revised as suggested.

-Figure 8 (a), (b), (d), (e): please specify the units of the color scale

Response: Revised as suggested.

-Page 23 Line 20: "underestimated"

Response: Revised as suggested.

**Reviewer #2**

This manuscript presents an evaluation of the WRF-GC model's performance in simulating the transport and transformation of atmospheric ozone in the Houston area during the TRACER-AQ campaign covering two ozone episodes in September 2021. The study established a model framework for atmospheric ozone in the Houston area and compared the modeled PBLH (as well as other meteorological parameters) and ozone profile to measurements, highlighting the limitations of ozone simulation and providing valuable insights for future ozone prediction. This aligns well with the scope of Geoscientific Model Development (GMD).

The manuscript is well-written, well-structured and well-referenced, drawing on relevant literature. The authors have conducted a rigorous analysis based on comprehensive observation data. The model is clearly described and observations are introduced in detail. However, some of the methods employed could benefit from further explanation to enhance the clarity of the research process and the presentation of some results seems to have room for improvement. This leads us to a more detailed discussion on specific scientific questions and issues raised by the study:

Response: We thank the reviewer for constructive comments and suggestions. We have taken them into account and made substantial updates to the corresponding texts and figures, which we believe have considerably improved the quality of our manuscript.

1. In Page 7 Line 40, simulations with various PBL schemes are chosen in the main analysis. Could you briefly explain why you are interested in the performance of PBL schemes most? [HRRR] is said to be the best, but according to Table S3 and S4, [Nudged] and [Re-init] have higher correlation coefficients and lower errors. Does the "best" mean the best among various meteorological drivers? Why [HRRR], instead of [Nudged] and [Re-init], is chosen in the main analysis?

Response: Thank you for bringing this to our attention. We have now incorporated further clarifications into P7 L38.

P7 L38: "*The WRF model generally reproduces observed temporal variability and spatial distribution in key meteorological parameters with a correlation coefficient higher than 0.5 in most cases. However, the model, regardless of configuration settings, shows persistent low biases in PBL heights, low biases in air temperatures, high biases in relative humidity, and high biases in wind speed (see Text S3 for details). While different WRF configuration has its own advantage in reducing model biases, [HRRR], [Nudged], and [Reinit] configurations stand out as the three best simulations based on campaign-wide statistics (see Text S3 for details). Considering that [Nudged] requires additional efforts to prepare observational datasets and [Reinit] needs to automate the model running process, [HRRR] is the easiest and the most effective option to reproduce meteorology for computationally expensive chemistry simulations and was thus selected to be presented in the analysis below. Meanwhile, the three simulations with different PBL schemes (i.e., [Base], [YSU], and [ACM2]) were also selected because the choice of the PBL scheme is crucial in determining PBL heights (Section 2.3.3), which is one of the major interests of this study. Therefore, we chose four simulations, that is [HRRR], [Base], [YSU], and [ACM2], in Table 1.*"

2. In Figure 3f, it is evident that the model misses the high wind speeds during the day and overestimates the wind speeds at night. However, the fitness (R) is similar to that in Figure 3e, which shows a higher degree of agreement. Is there a reasonable explanation for this? Are the correlation coefficients and NMBs in Figure 3 calculated by the diurnal mean (24 simulations vs 24 observations) or original records? If the former, are there differences between the bias and fitness

calculated from the original data and those presented in the study? Does taking the average of wind directions provide meaningful information?

Response: We understand the reviewer's concern and provide more clarification on the previous Figure 3e and 3f (now Figure 3c) below. Corresponding figures and texts have now been updated accordingly.

(1) In Figure 3f, it is evident that the model misses the high wind speeds during the day and overestimates the wind speeds at night. However, the fitness (R) is similar to that in Figure 3e, which shows a higher degree of agreement. Is there a reasonable explanation for this?

Figure R1 shows the analysis of wind speed during clean days vs. ozone episodes. Both (b) and (d) display a linear relationship indicated by blue trend lines. However, (d) is slightly more scattered while (b) is tighter around the trend line, indicating that (d) has a slightly lower correlation than (b). This is consistent with lower correlation during ozone episodes in Figures 3e than clean days in Figure 3f.

Four performance metrics are shown in subplots (b) and (d) to represent the bias and error. Mean Bias (MB) and Normalized Mean Bias (NMB) are lower during ozone episodes compared to clean days. This is because MB and NMB can compensate for positive and negative bias according to the formulas in Table R1. By taking take the absolute value or squares of the differences, Mean Absolute Error (MAE) and Root Mean Square Error (RMSE) are not compensated by positive and negative bias. Consequently, MAE and RMSE are larger during ozone episodes compared to clean days. This explains why MAE and RMSE exhibit larger values during ozone episodes compared to clean days in Figure 3e and 3f.

All statistics presented here are consistent with Figure 3e and 3f (now Figure 3c). We have now included RMSE along with R and NMB in Figure 3e and 3f (now Figure 3c) to present a more comprehensive set of statistics.

[Figure]

Figure R1. Diurnal variations of wind speed during (a) clean days versus (c) ozone episodes. Corresponding data points and statistics are plotted in scatterplots of (b) for clean days and (d) for ozone episodes.

Table R1. Model performance metrics used in this study. M is the model output, O is the observation, N is the number of samples.

| Performance Metrics | Formulas |
|---|---|
| Mean Bias (MB) | $MB = 1/N \sum_{i=1}^{N}(M_i - O_i)$ |
| Normalized Mean Bias (NMB) | $NMB = \dfrac{\sum_{i=1}^{N}(M_i - O_i)}{\sum_{i=1}^{N} O_i} \times 100\%$ |
| Mean Absolute Error (MAE) | $MAE = 1/N \sum_{i=1}^{N}|M_i - O_i|$ |
| Root Mean Square Error (RMSE) | $RMSE = \sqrt{1/N \sum_{i=1}^{N}(M_i - O_i)^2}$ |
| Correlation Coefficient (R) | $R = \dfrac{\sum_{i=1}^{N}(M_i - \overline{M})(O_i - \overline{O})}{\sqrt{\sum_{i=1}^{N}(M_i - \overline{M})^2}\sqrt{\sum_{i=1}^{N}(O_i - \overline{O})^2}}$ |

(2) Are the correlation coefficients and NMBs in Figure 3 calculated by the diurnal mean (24 simulations vs 24 observations) or original records? If the former, are there differences between the bias and fitness calculated from the original data and those presented in the study?

All statistics in Figure 3 were intended to be calculated using the diurnal mean over the original records. Figure 3 is intended to specifically represent diurnal variations only (especially for PBL height), instead of combination of diurnal and day-to-day variations that are present in the original records.

The choice of presenting diurnal variations is driven by the observed differences in diurnal variations of PBL heights in Section 3. We have now revised the main text to reflect this linkage more clearly in P10 L30 that "***Based on the observed differences in diurnal PBL variations between non-episode days and ozone episodes in Section 3, we first assessed the observation-model differences in diurnal variation of PBL heights and other meteorological variables in Section 4.1.*** ...".

Less performance is anticipated when using the original records over diurnal mean, since the former represents both diurnal and day-to-day variability and the latter only represents diurnal variability. Yet the relative performances between clean days versus ozone episodes are still consistent using both methods. For example, both methods show that the modeled PBL heights have less performance during ozone episodes than clean days (Table R2).

Table R2. Performance metrics using the diurnal mean versus the original records.

| | | Clean days | | Ozone episodes | |
|---|---|---|---|---|---|
| | | Diurnal mean | Original records | Diurnal mean | Original records |
| RH (%) | R | 0.99 | 0.85 | 0.91 | 0.8 |
| | NMB | 1% | 1% | -9% | -9% |
| | RMSE | 2.82 | 9.11 | 8.09 | 10.76 |
| PBL (km) | R | 0.88 | 0.49 | 0.46 | 0.15 |
| | NMB | -21% | -22% | -30% | -29% |
| | RMSE | 0.22 | 0.46 | 0.56 | 0.73 |

(3) Does taking the average of wind directions provide meaningful information?

The averages of wind directions (and also wind speeds) used in this paper are represented by vector averages, not scalar averages. Clarifications are now added in P9 L10-22 that "*The mean of wind speed and direction is calculated using the vector notation approach, a commonly used method in wind evaluations, as described in Yu et al. (2023). This method treats wind as vectors with their u (eastward) and v (northward) wind components. First, the mean u and v wind components are found by averaging all u and v wind values over a given time period. Then, the resultant vector is determined by taking the square root of the sum of the squares of the mean u and mean v wind components. The magnitude of resultant vector represents the mean wind speed, and the angle of the resultant vector represents the mean wind direction.*

*The difference between observed and modeled wind direction was calculated as below.*

$$\Delta = \begin{cases} M - O, & \text{when } |M - O| \leq 180° \\ (M - O)\left(1 - \dfrac{360}{|M - O|}\right), & \text{when } |M - O| > 180° \end{cases}$$

*where M is the model output, and O is the observation. The correlation between observed and modeled wind direction was determined by a circular correlation coefficient as below.*"

$$R = \frac{\sum_{i=1}^{N} \sin(M_i - \bar{M}) \sin(O_i - \bar{O})}{\sqrt{\sum_{i=1}^{N} \sin^2(M_i - \bar{M})} \sqrt{\sum_{i=1}^{N} \sin^2(O_i - \bar{O})}}$$

3. When a large amount of similar data is densely listed in the main text (e.g. Page 12 Line 7-14, Page 14 Line 14-21), is the presence of these data all necessary and supporting a particular conclusion? Is there a more concise and clear way to present the data instead of listing it?

Response: We agree with the reviewer's comment and have altered these paragraphs with substantial revisions to eliminate data listing. They now have a clearer presentation of the comparisons and associated conclusions.

P13 L20: "...*These comparisons demonstrate that the four model simulations generally underestimate the PBL height by 180–450 m throughout the day on non-episode days and by 10–760 m during the*

*daytime on ozone-episode days. Among the four simulations, [HRRR] best captures the observed mean height and decay rate during the daytime. Therefore, [HRRR] is selected to display its aerosol backscatter and potential temperature profiles in Figures 4 and 5, enabling further examination of its representation of the nighttime RL.*"

P15 L15-P16 L15: "*During ozone episodes, over land in the urban Houston region, the observed PBL heights gradually increase from 0.63±0.25 km in the morning (8:00-10:00 CDT), to 1.27±0.38 km at noon (11:00-13:00 CDT), and further to 1.69±0.23 km in the afternoon (14:00-16:00 CDT). Compared to land, the higher heat capacity in water leads to slower heating and cooling, resulting in a more stable atmosphere and shallower PBL. Over Galveston Bay, the observed heights are consistently lower by around 0.13-0.26 km during the three measured time periods.* Such daytime variation and land-water differences ...in most case during ozone episodes. **During non-episode days, the observed PBL height increases from 0.78±0.14 km in the morning to 1.07±0.24 km at noon over land, and slightly from 0.57±0.28 km in the morning to 0.65±0.34 km at noon over water. The model captures such variations during clean days less effectively, resulting in lower correlation and larger biases compared to ozone episodes (Table 2). One important reason for the lower model performance during clean days compared to ozone episodes is the substantial difference in the number of data points collected. There are significantly fewer data points available during the two clean days compared to the eight high ozone days (Table 2)*"

4. In Section 4.1, the evaluation of PBLH derived from the ceilometer uses the correlation coefficient R and NMB as indicators. However, in Section 4.2, when evaluating with mixed layer heights from HSRL-2, the metrics switch to Bias and RMS. Is the change in evaluation metrics necessary, and if so, what is the reason behind this?

Response: We agree with the reviewer's comment and have altered this section, as well as the entire manuscript, to consistently present correlation coefficient (R), normalized mean bias (NMB) and the root mean square error (RMSE).

5. In the comparison of ozone, the focus is mainly on bias rather than the correlation coefficient. Could this potentially lead to an insufficient evaluation of the simulation of temporal variations in ozone?

Response: Following the reviewer's suggestion, we have further added correlation coefficient on Figure 7, Figure 8, and Figure 9 (see below) and elaborated on these correlation coefficients in P23 L42.

P23 L42: "*To assess vertical variations below the first 4 km, we present performance metrics in Figure 7 for ozonesondes, Figure 8 for TROPOZ, and Figure 9 for HSRL-2. Different comparisons between observations and the model reflect distinct aspects. For instance, comparisons with ozonesonde pertain to vertical variations at a fixed location and time (R=0.46-0.77; NMB from -1% to -15%; RMSE=7-15 ppbv). This emphasis on a specific aspect explains why the correlation is higher compared to TROPOZ and HSRL-2, which encompass a broader range of variations. Comparisons with TROPOZ relate to vertical and temporal variations at a fixed location (R=0.18-0.39; NMB from -2% to 15%; RMSE=13-17 ppbv). Comparisons with HSRL-2 represents a combination of vertical, temporal, and spatial variations (R=0.18-0.76; NMB from -7% to 5%; RMSE=7-13 ppbv). The above statistics exclude one or two extreme cases in each observation. Despite the differences in correlation resulting from the diverse representations of variations, biases are similar when compared to the three different observations.*"

[Figure]

Figure 7. Vertical profiles of (a) potential temperature and (b) ozone from ozonesonde measurements and the WRF-GC [HRRR] simulation at La Porte during September 8-11 and September 23-26.

[Figure]

Figure 8. Time series of the vertical ozone profile from the TROPOZ ozone lidar (a, d) and the WRF-GC [HRRR] simulation (b, e) at La Porte. Observed and modeled boundary layer heights are inserted, respectively. Dots represent the modeled residual layer identified in this study. Line plots (c, f) show ozone differences (model minus observation) at the free troposphere (2-3 km) and the boundary layer (0.5-1 km) from the TROPOZ as well as the near-ground (< 50m) from the model 49i ozone analyzer.

[Figure]

Figure 9. Vertical ozone profiles from (a, d) the HSRL-2 and (b, e) the WRF-GC [HRRR] simulation. The profiles are taken from a flight track (Fig.1) over urban Houston and Galveston Bay at around 11:00-13:00 CDT each day. Line plots (c, f) show ozone differences (model minus observation) at the free troposphere (2-3 km) and the boundary layer (0.5-1 km).

6.  In Figure S2, only wind directions are shown with correlation coefficients. Are they considered as continuous variables? For example, are 0° and 359° treated as match or mismatch? Could you also include one of the performance index (e.g. R) in the figures of other variables?

Response: Following the reviewer's suggestion, we have further added correlation coefficients for all variables in Figure S2.

Wind directions are not considered as continuous variables. Instead, they are considered as vectors and all statistics related to wind are conducted using vector calculations instead of scalar calculations. Please kindly refer to responses to the review's comment # 2 above for full descriptions. The corresponding clarifications are now added both in the main text in P9 L10 as well as in the supplement in P3 L18.

P3 L18 in the supplement: "*The mean of wind speed and direction is calculated using the vector notation approach, a commonly used method in wind evaluations, as described in Yu et al. (2023). This method treats wind as vectors with their u (eastward) and v (northward) wind components. First, the mean u and v wind components are found by averaging all u and v wind values over a given time period. Then, the resultant vector is determined by taking the square root of the sum of the squares of the mean u and mean v wind components. The magnitude of resultant vector represents the mean wind speed, and the angle of the resultant vector represents the mean wind direction.*

*The difference between observed and modeled wind direction was calculated as below.*

$$\Delta = \begin{cases} M - O, & \text{when } |M - O| \le 180° \\ (M - O)\left(1 - \dfrac{360}{|M - O|}\right), & \text{when } |M - O| > 180° \end{cases}$$

*where M is the model output, and O is the observation. The correlation between observed and modeled wind direction was determined by a circular correlation coefficient as below.*"

$$R = \frac{\sum_{i=1}^{N} \sin(M_i - \bar{M})\sin(O_i - \bar{O})}{\sqrt{\sum_{i=1}^{N} \sin^2(M_i - \bar{M})} \sqrt{\sum_{i=1}^{N} \sin^2(O_i - \bar{O})}}$$

7.  Again, in Figure S3d and S5d, wind direction is shown in the same way as the other parameters, which results in some small differences, such as between 0° and 359°, appearing large in the figure.

Response: We have revised Figures S3 and S5 (including Figure S3d and S5d on wind direction) to display the observation-model differences (i.e., model minus observation). In this way, near 0 on the y-axis implies small observation-model differences, while far from 0 indicates large differences.

8.  Text S4 and Figure S6 shows a group of observations. What conclusions did you make from them? Why are they presented in this paper? Have they been compared with the modeled profile, as TROPOZ?

Response: We agree with the reviewer's comment and have removed Text S4 and Figure S6.

9.  We have known from this study the performance and limitations of the modeling system, but I would like to suggest that the paper is heavy on conclusions but light on discussion. I recommend expanding the discussion to provide more context and interpretation of the findings.

Response: We agree with the reviewer's comment and have further expanded the implication section in P26 L19 and P26 L46.

P26 L19: "*Based on these evaluations, we summarized model limitations that prevent a more accurate simulation of PBL heights and the vertical ozone distribution during TRACER-AQ. The first limitation is the single-layer PBL representation. The WRF model only diagnoses the SBL at night, despite the model simulating different physical and thermodynamic properties of multiple nocturnal layers above the SBL. For example, the RL is not identified by the model as a standard diagnosis; this prevents the direct comparison of the model outputs with the observed RL at night.* **Further efforts are needed to identify and incorporate the RL into the model's standard outputs. Alternative modules aimed at identifying the PBL using simulated vertical backscatter gradients can also enhance the validation of PBL heights with backscatter-derived observations.**"

P26 L33: "*Our findings have implications for the predictivity of ozone's vertical mixing and distribution across different modeling systems. For example, WRF is widely used in various meteorology-chemistry coupling systems with different treatments of boundary layer mixing. In WRF-Chem, boundary layer mixing in the chemistry part uses a mixing coefficient originating in WRF such that the boundary layer mixing calculations in the meteorology and chemistry parts share the same set of coefficients. In WRF-GC, the chemistry part from GEOS-Chem only takes the PBL height from WRF as the maximum height for boundary layer mixing but conducts independent calculations of boundary layer mixing using its own internal methods, which are not reliant on WRF. Unlike online coupled WRF-Chem and WRF-GC, WRF is offline coupled to CAMx in the WRF-CAMx system, and the boundary layer mixing in the chemistry part of CAMx is subject to WRF output frequency instead of the native transport time step in WRF.* **Considering these distinct treatments of boundary layer mixing in models, the single-layer PBL representation can have varying impacts on the simulation of vertical mixing and, consequently, the vertical distribution of ozone and other air pollutants. Thus, it is essential to understand the differences in boundary layer mixing among different meteorology-chemistry coupling systems. Follow-up studies to this work will address these aspects with a detailed analysis of vertical mixing processes in various models.**"

Some technical corrections or typos:

Page 6 Figure 1: Font sizes of the latitude and longitude labels are inconsistent. The labels on the left subfigures are smaller and blurry, which affects readability.

Response: Revised as suggested.

Page 8 Line 17: "MNYY" should be MYNN

Response: Thank you. We have corrected this.

Page 9 Figure 2: If the colorbars for the three subfigures are identical, there is no need to display them three times. Having separate colorbars for each figure could lead to the misconception that the scales are different for each one.

Response: Revised as suggested.

Page 13 Figure 4-5: Ensure consistency in subheadings, y-axis titles, etc. What is the difference between the y-axis "Height" and "Altitude"? Ensure that the text on the figures is clear and legible.

Response: The figures have been fixed.

Figures S3 and S5: the overlapping lines hinder readability.

Response: We have revised Figures S3 and S5 to display the observation-model differences (i.e., model minus observation). This alteration allows for improved visualization of the diverse performance of multiple model configurations, as well as better identification of dots with different colors, as opposed to the lines used in the previous version.

In general, the quality of the figures could be improved. The resolution is too low or the font size is too small in some figures, which affects readability, especially in the Supplementary Information (e.g., Figure S2). There is a lack of consistency in the font and font size, as well as the titles and labels across subfigures. Some figures are missing necessary axis titles or colorbar descriptions, and some figures have subfigures that are not aligned.

Response: Following the reviewer's suggestion, we have made updates to every single figure in the manuscript as well as the supplement to improve legibility.

---

## Author Response (AR2)

Editor comments:
Congratulations, you are nearly there. I have only two minor technical comment that should be fixed before accepting the manuscript for publication in GMD. Following the guidelines of GMD, the Zenodo link with the frozen version of the model used should be moved to the reference list following Haipeng Lin. (2020). jimmielin/wrf-gc-pt2-paper-code-nested: WRF-GC with nested functionality - for paper submission (v3.0). Zenodo. https://doi.org/10.5281/zenodo.4395258, and simply having Haipeng Lin (2020) on the code availability text. Then you do not need the date of reference either. The same applies also to the Zenodo link in Data availability section. Please fix these two minor technical points.

Author response:
Thank you Prof. Järvi for your kind reminder. These two are now revised as suggested.